

# The onset of chaos in nonautonomous dissipative dynamical systems: A low-order ocean–model case study[*]

Stefano Pierini[1,2], Mickaël D. Chekroun[3], and Michael Ghil[3,4]

[1]Universita' di Napoli Parthenope, Naples, Italy
[2]CoNISMa, Rome, Italy
[3] University of California at Los Angeles, Los Angeles, California, USA
[4]Ecole Normale Supérieure and PSL Research University, Paris, France

*Correspondence to:* Stefano Pierini (stefano.pierini@uniparthenope.it)

**Abstract.** The transition to chaos induced by periodic forcing in systems that are not chaotic in the autonomous limit is studied with a four-dimensional nonlinear spectral ocean model. The analysis is based on the systematic construction of the system's pullback attractors (PBAs) through ensemble simulations derived from a large number of initial states in the remote past. A preliminary analysis of the autonomous system is carried out by constructing its bifurcation diagram, as well as by calculating

a metric measuring the mean distance between two initially nearby trajectories, along with the system's entropy. We find that nonchaotic attractors can still exhibit sensitive dependence on initial data; this apparent paradox is resolved by noting that the dependence only concerns the phase of the periodic trajectories, and that it disappears once the latter have converged onto the attractor. The periodically forced system, analyzed by the same methods, yields periodic or chaotic PBAs depending on the periodic forcing's amplitude $\varepsilon$. A new diagnostic method — based on the cross-correlation between two initially nearby

trajectories — is proposed to characterize the transition between the two types of behavior. Transition to chaos is found to occur abruptly at a critical value $\varepsilon_\mathrm{c}$ and begins with the intermittent emergence of periodic oscillations with distinct phases. The same diagnostic method is finally shown to be a useful tool for autonomous and aperiodically forced systems as well.

## 1 Introduction and motivation

Understanding the mechanisms that lead to the onset of chaos in dissipative dynamical systems is of fundamental importance

both from a cognitive viewpoint and for a correct use of the mathematical models on which the systems are based. Chaos arises in such systems as a control parameter in the governing equations crosses a given threshold. A huge amount of work has been devoted to analyzing the transition to chaos in the framework of autonomous dynamical systems, i.e., in systems in which the external forcing and the coefficients do not depend on time. The various routes to chaos in autonomous dissipative systems include period doubling cascade, intermittency and crisis, quasi-periodic routes, global bifurcations (e.g., Strogatz,

1994; Nicolis, 1995; Hilborn, 2000; Ott, 2002; Tél and Gruiz, 2006).

[*]This paper is dedicated to the memory of Anna Trevisan and to her contributions to the applications of dynamical systems theory to the climate sciences.



Nonautonomous dissipative dynamical systems represent a crucial extension of autonomous systems for practical applications, since the external forcing in most real systems — whether deterministic, random or both — depends, typically, on time. Despite their importance, nonautonomous systems have received, until recently, less attention than autonomous systems. Transition to chaos induced by time-dependent forcing has, nonetheless, been studied in several significant cases. A classical example is the Van der Pol oscillator (Van der Pol, 1920, 1926), in which chaotic relaxation oscillations emerge under the effect of an external periodic forcing. A few more recent examples in the climate sciences include (i) transition to chaos due to quasi-periodic forcing (Le Treut and Ghil, 1983; Ghil, 1994); (ii) modification of the autonomous transition by periodic forcing (Sushama et al., 2007); and (iii) the important contributions of Anna Trevisan and colleagues to the data assimilation problem for chaotic systems, in which the data stream can be seen essentially as a time-dependent forcing (Trevisan and Uboldi, 2004; Carrassi et al., 2008a, b; Trevisan and Palatella, 2011).

The onset of chaos is analyzed here in the framework of nonautonomous systems, which has recently received rapidly increasing attention in the context of climate dynamics (Ghil et al., 2008; Chekroun et al., 2011; Bódai and Tél, 2012; Bódai et al., 2013; Pierini, 2014; Drótos et al., 2015, 2017; Ghil, 2015, 2017; Pierini et al., 2016; Lucarini et al., 2017). Our study focuses on a four-dimensional nonlinear spectral ocean model (Pierini, 2011), which is subjected to periodic forcing, chosen as the simplest form of time dependence. Cases will be considered that are nonchaotic in the autonomous limit, so that the chaos that emerges in the system is strictly associated with the nonstationarity of the forcing.

The study makes use of ensemble simulations performed with many initial states distributed in a given subset of phase space, following the methodology of Pierini (2014) and Pierini et al. (2016). The overall idea is that the relevant information in the climate system must be derived from statistical analyses of an ensemble of different system trajectories, each corresponding to a different initial state, provided that the corresponding trajectories have converged to the system's time-dependent attractor.

Such an attractor is called a pullback attractor (PBA; e.g., Ghil et al., 2008; Chekroun et al., 2011; Kloeden and Rasmussen, 2011; Carvalho et al., 2012) in the mathematical literature and a snapshot attractor (e.g., Romeiras et al., 1990; Bódai and Tél, 2012; Bódai et al., 2013) in the physical literature; it provides the natural extension to nonautonomous dissipative dynamical systems of the classical concept of an attractor that is fixed in time for autonomous systems. A global PBA is defined as a time-dependent set $\mathcal{A}(t)$ in the system's phase space that is invariant under its governing equations, along with the equally time-dependent, invariant measure $\mu(t)$ supported on this set, and to which all trajectories starting in the remote past converge (Arnold, 1998; Rasmussen, 2007; Kloeden and Rasmussen, 2011; Carvalho et al., 2012). In the deterministic case, it is understood that $\mathcal{A}(t)$ depends also on the particular forcing, say $F(t)$, that is being applied, but this dependence is usually not kept track of in the notation. In the random case, the PBA is called a random attractor, and the dependence on the specific realization $\omega$ of the noise process is often included in the notation, as $\mathcal{A}(t, \omega)$.

Pierini et al. (2016) rigourously proved that a weakly dissipative nonlinear model like the one used there and herein does possess a global PBA, subject to mild integrability conditions on the forcing. In the present study, the numerical approach used for the systematic investigation of the system's PBAs follows Pierini (2014) and Pierini et al. (2016). Further diagnostic tools will be introduced for the present periodic-forcing set-up, and a new diagnostic tool will also be proposed to monitor the onset of chaos in our nonautonomous system.





The paper is organized as follows. In Sect. 2, the mathematical model is described. In Sect. 3, the main properties of the autonomous system are summarized and an apparent paradox related to the sensitivity to initial states in the periodic regime is discussed. In Sect. 4, the results obtained for the periodically forced system are presented and discussed; the new cross-correlation–based method specifically formulated to characterize the onset of chaos is introduced and applied to the specific

case at hand. This method helps characterize the transition to chaos as the amplitude of the periodic forcing increases, as well as document the coexistence of local PBAs with chaotic and nonchaotic behavior within the model's global PBA. In Sect. 5, the same method is shown to be a useful tool also for autonomous systems and even for some aperiodically forced ones. Finally, in section 6 the results are summarized and conclusions are drawn. An appendix illustrates in greater detail the coexistence of local PBAs that are chaotic and not chaotic in the setting of a periodically forced Van der Pol–Duffing oscillator.

## 2 Model description

The highly idealized model of the oceans' wind-driven, double-gyre circulation (Ghil, 2017, and references therein) used in the present study is governed by the system of four nonlinear, coupled ordinary differential equations derived by Pierini (2011). This author introduced such a low-order model to complement the process studies on the Kuroshio Extension's low-frequency variability previously carried out with a much more detailed, primitive equation ocean model (e.g., Pierini, 2006; Pierini et al.,

2009; Pierini and Dijkstra, 2009). The same low-order model was later used by Pierini (2014) and Pierini et al. (2016) to explore the PBAs of the system in various cases. Here we merely review the main aspects of the model; for all the technical details and parameter values, the interested reader should kindly refer to Pierini (2011).

The dynamics is governed by the evolution equation of potential vorticity in the quasigeostrophic approximation on the beta-plane for a shallow layer of fluid, superimposed on an infinitely deep quiescent lower layer. Pierini (2006) found such a reduced-

gravity model to be a good approximation for process studies of the Kuroshio Extension's low-frequency variability (LFV). The flow is described by the streamfunction $\psi(\mathbf{x},t)$: like in the previous studies, $\psi$, the horizontal coordinates $\mathbf{x}=(x,y)$ and the time $t$ are dimensionless, but the dimensional time will be plotted in all the time series presented in this study to emphasize the typical time scales of the oceanic phenomena under investigation.

A four-dimensional spectral model is obtained by expanding the streamfunction in a rectangular domain as follows:

$$\psi(\mathbf{x},t) = \sum_{i=1}^{4} \Psi_i(t)\,|i\rangle. \tag{1}$$

The orthonormal basis $|i\rangle$ is defined as:

$$|1\rangle = e^{-\alpha x}\sin x\sin y; \quad |2\rangle = e^{-\alpha x}\sin x\sin 2y; \quad |3\rangle = e^{-\alpha x}\sin 2x\sin y; \quad |4\rangle = e^{-\alpha x}\sin 2x\sin 2y$$

where $\alpha$ is a real positive constant. This basis satisfies the free-slip boundary conditions along the borders of the rectangular domain; it also captures the oceanic flow's westward intensification thanks to the exponential factor first introduced in the

two-dimensional model of Jiang et al. (1995).

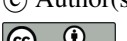



The four nonlinear coupled ordinary differential equations that govern the evolution of the vector $\boldsymbol{\Psi}(t) = (\Psi_1, \Psi_2, \Psi_3, \Psi_4)$ can be written, cf. Pierini (2011), as

$$\frac{\mathrm{d}\boldsymbol{\Psi}}{\mathrm{d}t} + \boldsymbol{\Psi}\mathbf{J}\boldsymbol{\Psi} + \mathbf{L}\boldsymbol{\Psi} = G(t)\,\mathbf{w}; \tag{2}$$

The coefficients of the nonlinear and linear terms in the equation are encapsulated by the rank-3 and rank-2 tensors $\mathbf{J}$ and

$\mathbf{L}$, respectively, and the forcing is represented by the vector $\mathbf{w}$, cf. Pierini (2011). The forcing $\mathbf{w}$ is obtained from a suitable double-gyre surface wind stress curl, while $G$ is defined in the present paper to be periodic,

$$G(t) = \gamma\left[1 + \varepsilon\sin(\omega t)\right], \tag{3}$$

with period $T_{\mathrm{p}} = 2\pi/\omega$, while $\gamma$ and $\varepsilon$ are positive dimensionless parameters.

To construct the system's PBAs, ensembles of forward time integrations are carried out; each of these starts at $t = 0$ from a

different initial point contained in a given subset $\Omega$ of the model's four-dimensional phase space, and ends at $t = T_* = 400$ yr. Following Pierini (2014) and Pierini et al. (2016), the four-dimensional hypercube $\Omega$ is defined as:

$$\Psi_1, \Psi_2 \in [-70, 150]; \quad \Psi_3, \Psi_4 \in [-150, 120] \tag{4}$$

and the initial data are all chosen to satisfy $\Psi_1 = \Psi_2$ and $\Psi_3 = \Psi_4$, i.e., to lie within a plane set embedded in $\Omega$.

The ensembles will consist of 15 000 initial data at $t = 0$ that are regularly spaced either in $\Omega$ or in a small subset thereof. For

the sake of graphical representation, maps of various quantities will be plotted in the rectangle $\Gamma \equiv \{-70 \leq \Psi_1 \leq 150, -150 \leq \Psi_3 \leq 120\} \subset \Omega$ that lies in the $(\Psi_1, \Psi_3)$-plane. In the discussion of the results, we will refer, for the sake of simplicity and concision, to the model's trajectories as being defined in the $(\Psi_1, \Psi_3)$-plane but, naturally, the actual trajectories evolve in the full four-dimensional phase space.

## 3  The autonomous system

### 3.1  The autonomous model's attractors

We begin by analyzing some basic properties of the autonomous system that will be useful in the subsequent investigation. The bifurcation diagram of Fig. 1a shows the range of variability of $\Psi_1$ vs. the forcing parameter $\gamma$. The value $\gamma = 1$ corresponds to a global bifurcation that manifests itself by a sudden transition from a small-amplitude limit cycle to a relaxation oscillation with a much higher amplitude. The previous results in this respect (Pierini, 2011; Pierini et al., 2016) will be further bolstered

by those in Sect. 5 (Figs. 14 and 15) herein that are based on the diagnostic tool proposed in Sect. 4.2.

Figure 1b shows the limit cycle in $\Gamma$ arising from arbitrary initial data for $\gamma = 1.1$, which corresponds to the red vertical line in panel (a). For $\gamma = 1.35$ (green line in panel (a)) the attractor is chaotic. In this case, the map of the suitably normalized decimal logarithm of the probability density function (PDF) of the trajectories in $\Gamma$ is plotted in Fig. 1c.

In an autonomous dynamical system, the attractors do, by definition, not depend on time, i.e., an attractor is a geometric

object in phase space that is fixed in time. However, any attractor that is not a fixed point — whether a limit cycle, torus or





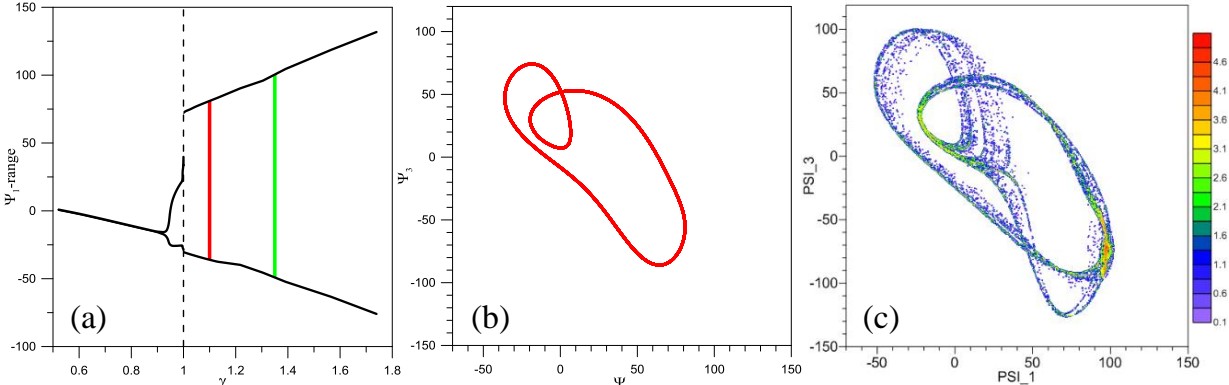

**Figure 1.** Behavior of the autonomous ocean model, for which $\varepsilon = 0$ in Eq. (3). (a) Bifurcation diagram, in which the range of the variable $\Psi_1$ is plotted vs. the wind stress intensity $\gamma$; the two cases $\gamma = 1.1$ and $1.35$ discussed in the text are indicated with a red and a green vertical line, respectively. (b) Limit cycle in the $(\Psi_1, \Psi_3)$-plane, plotted after spinup, that arises from $\Gamma$ for $\gamma = 1.1$. (c) Normalized logarithm of the PDF of localization of the trajectories in the $(\Psi_1, \Psi_3)$-plane for the chaotic case $\gamma = 1.35$, using 15 000 trajectories regularly spaced in $\Gamma$ at $t = 0$; see text for details.

strange attractor — can contain time-dependent trajectories. Such ensembles of trajectories arising from specific sets of initial states will be plotted to illustrate the attractors of the autonomous system studied herein.

Following Pierini et al. (2016), in Figs. 2(a,b) the attractors that correspond to the two cases in Figs. 1(b,c) are represented by the time series of $P_{\Psi_3}(t) = \log_{10}(1000 p_{\Psi_3})$, where $p_{\Psi_3}(t)$ is the PDF of localization of the $\Psi_3$-variable; see Pierini et al. (2016) for technical details. The dense distribution of $P_{\Psi_3}(t)$ for $\gamma = 1.35$, as seen in Fig. 2b, is clearly associated with the chaotic character of the flow, while the periodic distribution of $P_{\Psi_3}(t)$ that corresponds to $\gamma = 1.1$ in Fig. 2a is due to the different phases that each trajectory attains on the limit cycle, depending on $(X, Y)$.

In Figs. 2(c,d), the same attractors are characterized through the metric $\sigma$ that was introduced by Pierini et al. (2016); this metric measures the mean divergence of trajectories over the total integration time $T_*$, and is defined as follows: The instantaneous Euclidean distance between two initially close trajectories is $\delta(t)$, and its normalized value is given by $\delta_{\mathrm{n}}(t) = \delta(t)/\delta(0)$. Then $\sigma$ is simply the average of $\delta_{\mathrm{n}}$ over $T_*$,

$$\sigma(X, Y) = \frac{1}{T_*} \int_0^{T_*} \delta_{\mathrm{n}}(t)\mathrm{d}t, \quad \text{with} \quad (X, Y) \equiv (\Psi_1(0), \Psi_3(0)) \in \Gamma. \tag{5}$$

Pierini et al. (2016) found the quantity $\sigma$ to be a good indicator of the degree of sensitivity of the system's evolution with respect to the initial state during the phase of convergence to the attractor.

The map of $\sigma$ in Fig. 2c reveals, in the autonomous case at hand, the same striking features found by Pierini et al. (2016) for the nonautonomous, aperiodic-forcing case, namely the coexistence of extended regions of $\Gamma$ with $\sigma \leqslant 1$, shown by cold colors, and with $\sigma > 1$, appearing as warm colors. In the first case, two trajectories that are initially close remain close at all





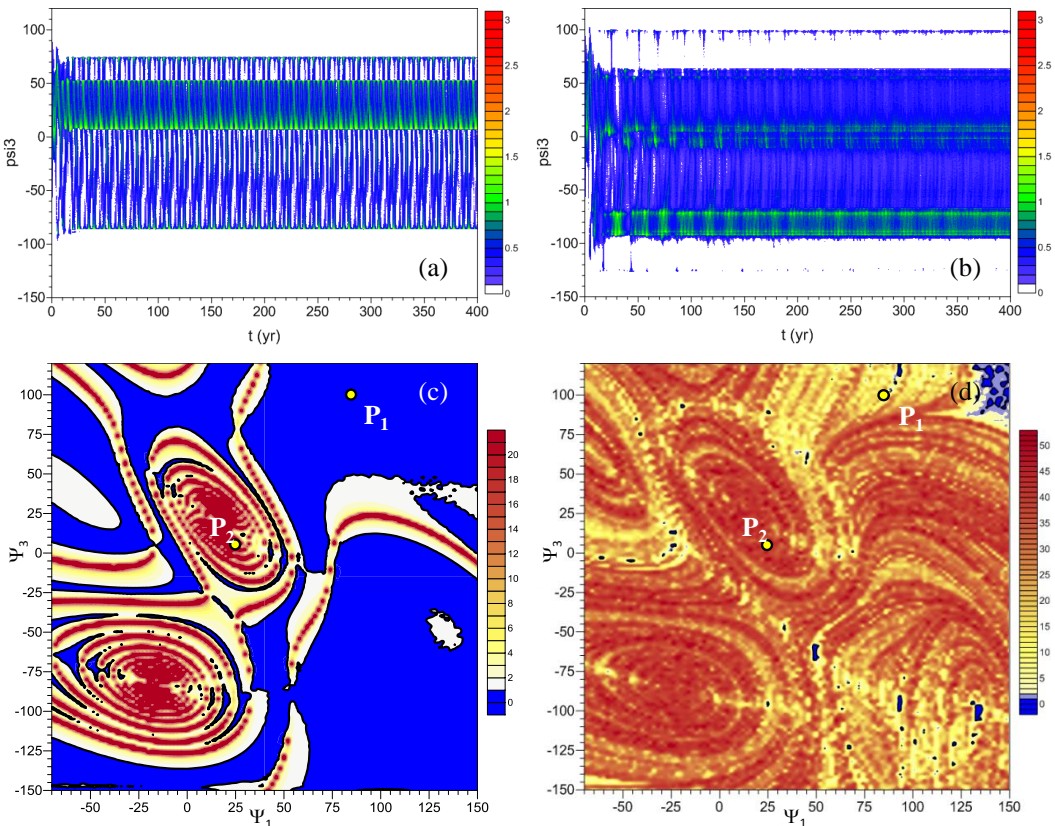

**Figure 2.** Distinct autonomous regime behavior for (a,c) $\gamma = 1.1$, and (b,d) for $\gamma = 1.35$. Upper panels: time series of $P_{\Psi_3}$ for (a) $\gamma = 1.1$, and (b) $\gamma = 1.35$. Lower panels: maps of the mean normalized distance $\sigma$ for (c) $\gamma = 1.1$, and (d) for $\gamma = 1.35$; the points $P_1 = (85, 100)$ and $P_2 = (25, 5)$ appear in the panels (c) and (d), respectively. Note the different scales in the two maps; 15 000 trajectories, with regularly spaced initial points in $\Gamma$, were used for both maps.

times, as seen in Fig. 3a. In the second case, though, two trajectories that are initially close may attain a large phase difference once they have converged to the attractor, cf. Fig. 3b, while still remaining perfectly coherent.

In the chaotic case $\gamma = 1.35$, the warm-color regions, in which $\sigma > 1$ overwhelm the cold-color regions, in which $\sigma \leq 1$, cf. Fig. 2d. To illustrate the two types of behavior, Figs. 3(c,d) show the evolution of $\Psi_3(t)$ of two initially nearby trajectories: if

5   $\sigma > 1$ initially, as is the case near point $P_2$, the two aperiodic signals become incoherent (Fig. 3d). Finally, it is worth noting that, for simulations with sufficiently small $\gamma$ (not shown), $\sigma < 1$ everywhere.

A different and useful way of looking at these two types of behavior is to analyze the corresponding mixing properties of the flow in the model's phase space. To do so, one can make use of the system's entropy (Shannon, 1948):

$$S_\vartheta(t) = -\sum_{k=1}^{N} p_k \ln p_k. \qquad (6)$$

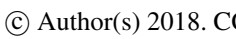


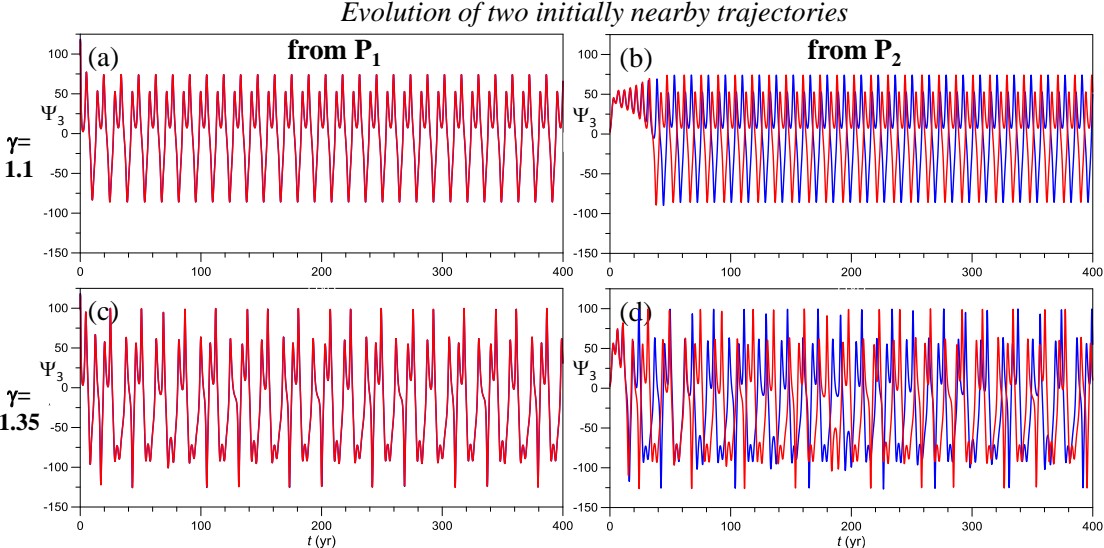

**Figure 3.** Typical behavior of time series of $\Psi_3(t)$ for different values of the parameter $\gamma$ and different initial points in $\Gamma$. (a) Two trajectories obtained for $\gamma = 1.1$ and initialized at $P_1$ (red line) and at a nearby point (blue line). (b) Same as in panel (a), but for two trajectories starting from the point $P_2$ (red) and near it (blue). (c,d) Same as in (a,b) but for $\gamma = 1.35$.

Here $\Gamma$ is decomposed into a regular grid of $N$ square cells of width $\Delta\Psi$ — with $N = 15\,000$ and $\Delta\Psi = 2$, so that the grid corresponds to that of the initial data used in our ensemble simulations — and $p_k(t)$ is the probability of localization in the $k$-th cell at time $t$ of the trajectories emanating at time $t = 0$ from a given subset $\vartheta \subset \Gamma$.

Figure 4a shows the intersection with the $(\Psi_1, \Psi_3)$-plane at $t = 400$ yr of 15 000 trajectories originating from the box $\vartheta_1$

that coincides with the $\Delta\Psi \times \Delta\Psi$ grid cell centered at $P_1$; the red dots correspond to the case $\gamma = 1.1$ and the green dots to $\gamma = 1.35$. Figure 4b shows $S_{\vartheta_1}$ for the two cases; note that $S_{\vartheta_1}(0) = 0$ since all the initial states lie in the single cell $\vartheta_1$, and thus $p_1 = 1$. The entropy of the periodic case $\gamma = 1.1$, characterized initially by $\sigma < 1$, oscillates between 0 and 1, with the final evolution limited to virtually a single cell over the limit cycle; on the contrary, the chaotic case leads to the trajectories' being scattered over the chaotic attractor, with the entropy (green line in Fig. 4b) reaching large values.

Figures 4(c,d) show the same quantities for the initial $\Delta\Psi \times \Delta\Psi$ box $\vartheta_2$ centered at $P_2$. The chaotic case is similar to that for $\vartheta_1$, but with a greater entropy; on the other hand, the periodic case differs in that now $\sigma > 1$ initially, cf. Fig. 2c. Figure 4c shows that the asymptotic evolution of the very small $\vartheta_2$ covers a limited but significant part of the limit cycle, as seen by comparing this figure with Fig. 1b; the corresponding entropy in Fig. 4d eventually oscillates periodically between the values $S_{\vartheta_2} \sim 2.2 - 3.7$. Figure 4 thus demonstrates clearly the usefulness of the metric $\sigma$ in characterizing subsets of $\Gamma$ and the effect

of the control parameter $\gamma$.





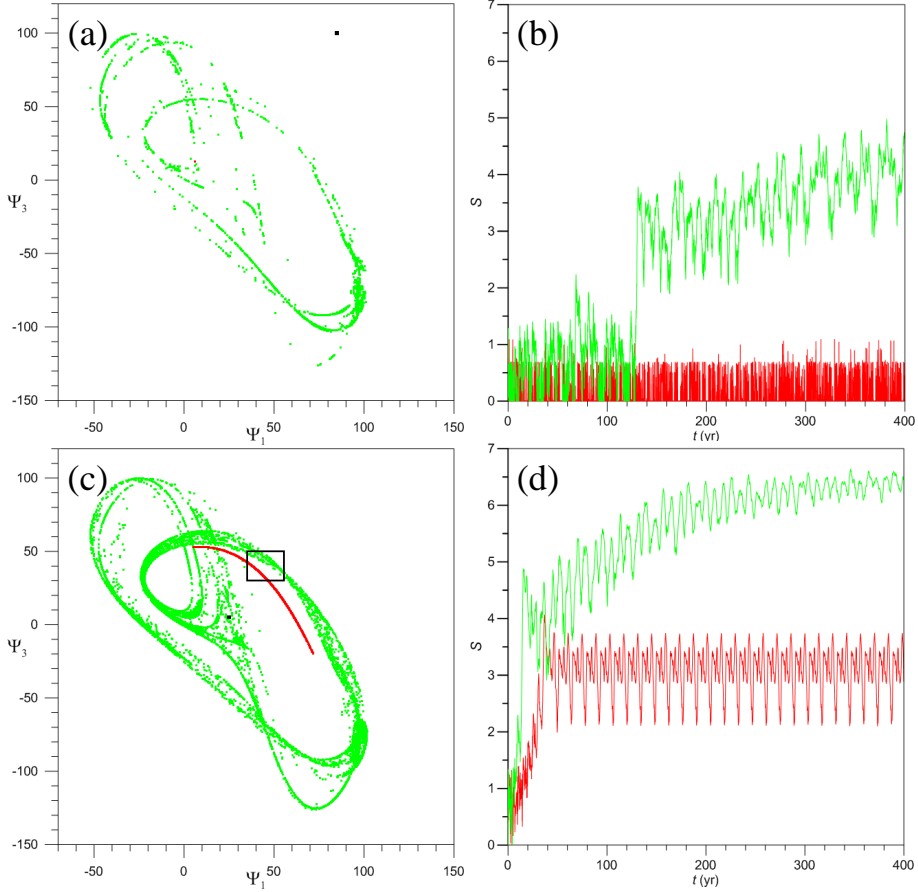

**Figure 4.** Chaotic and not chaotic behavior of the autonomous model, for time-independent forcing intensity $\gamma = 1.1$ (red) and $\gamma = 1.35$ (green), respectively. Typical behavior of (a,c) the trajectories in the model's phase plane $(\Psi_1, \Psi_3)$; and (b,d) of the model's entropy $S_\vartheta(t)$. (a) Intersection with the $(\Psi_1, \Psi_3)$-plane at $t = 300$ yr of 15 000 trajectories emanating from the small square box $\vartheta_1$ of width $\Delta\Psi = 2$ and centered at the point $P_1$ (black dot), for $\gamma = 1.1$ (red dots) and for $\gamma = 1.35$ (green dots). (b) Time series of the corresponding entropy $S_{\vartheta_1}$ for $\gamma = 1.1$ (red line) and $\gamma = 1.35$ (green line). (c,d) Same as panels (a,c), but for the initial box $\vartheta_2$ centered at $P_2$, likewise shown as a black dot in (c). For the evolution of the points contained at $t = 300$ yr in the black rectangle of panel (c), see Fig. 5 below.

## 3.2 An apparent paradox

We conclude the analysis of the autonomous system by discussing an apparent paradox. We have just seen that, in regions of $\Gamma$ where $\sigma > 1$, the trajectories for $\gamma = 1.1$ exhibit very sensitive dependence on initial data, as shown, for instance, by the red dots in Fig. 4c and by the red line in Fig. 4d. This property is usually associated with chaotic dynamics, but in this case the dynamics is periodic. This paradox is resolved by noting that the sensitive dependence occurs while the initial data are distributed over all of $\Gamma$; once a trajectory has converged onto the attractor, the sensitivity of the dependence disappears, as



shown below. On the contrary, for $\gamma = 1.35$, the highly sensitive dependence on the initial data always holds, on the attractor as well as off it, in excellent agreement with the chaotic character of the dynamics in this case.

We must show, therefore, that the trajectories are stable for $\gamma = 1.1$ and unstable for $\gamma = 1.35$, once they have settled onto the attractor. This distinction between the two cases can already be inferred from Figs. 3 and 4 but it is worth investigating the

issue in greater detail. The usual quantitative approach relies on the computation of the mean finite-time Lyapunov exponent $\lambda$ of each trajectory (e.g., Ott, 2002).

The results (not shown) prove unequivocally the assumption above, but the exponents are highly dependent (i) on the time $T_\lambda$ over which the finite-time exponents are computed; and (ii) on the amplitude of the perturbation superimposed on the reference trajectory at each time step $T_\lambda$. Moreover, the assumption of exponential divergence of chaotic trajectories is not fully met in

our highly nonlinear framework, so that the transition between periodic and chaotic dynamics may actually occur at a value of $\lambda$ that is not exactly equal to 0. A qualitative but more robust method is instead summarized in Fig. 5, and we propose, furthermore, in Sect. 4 an alternative quantitative method.

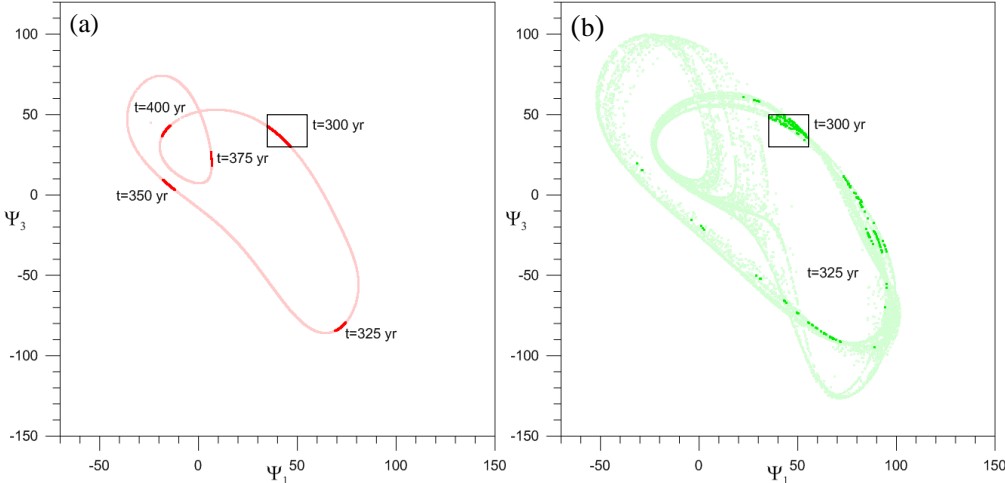

**Figure 5.** Chaotic and not chaotic behavior of the autonomous model. (a) Evolution of the 1774 points (red dots) contained at $t = 300$ yr in the black rectangle of Fig. 4c for $\gamma = 1.1$; the attractor is illustrated by the entire set of 15 000 points, shown in light red. (b) Same but for the 135 points (green dots) that lie within the same rectangle for $\gamma = 1.35$; in this case, the attractor composed of the 15 000 points is shown in light green. Due to the chaotic nature of the dynamics in this case, only one snapshot, after 25 yr, is drawn.

Let us consider the points lying in the black rectangle shown in Fig. 4c at $t = 300$ yr: the corresponding evolution at four subsequent time instants, $T_\Delta = 25$ years apart, is shown in Fig. 5a for $\gamma = 1.1$ (red dots). Note that the period of the orbits on

the attractor is $T_{p^*} = 14.08$ yr, i.e. $T_\Delta > T_{p^*}$.

The stability of the trajectories under consideration in Fig. 5a is clearly demonstrated by the compact form and limited extent of the cluster of points in its evolution, which spans the attractor roughly six times in $4\Delta = 100$ yr. On the contrary, Fig. 5b





shows that for $\gamma = 1.35$ (green dots) the compact form of the initial cluster is lost already after a single 25-yr lapse of time. The trajectories will thus soon be scattered over the strange attractor, due to their divergence.

In conclusion, our autonomous system becomes chaotic for sufficiently large values of $\gamma$, e.g., for $\gamma = 1.35$. The system's periodic regime spans a range of $\gamma$ that includes the bifurcation at $\gamma = 1$, which is apparent in Fig. 1a.

We have shown, moreover, when $\gamma = 1.1$, that regions of $\Gamma$ exist within which the mean normalized distance $\sigma$ between two initially nearby trajectories is larger than unity; see again Fig. 2c. In this case, despite the attractor's being a limit cycle, the trajectories leaving from such regions of phase space experience sensitive dependence on the initial data. This property, typically associated with chaotic systems, is not in contradiction with the periodic character of the solutions: as a matter of fact, the trajectories under discussion are stable and the sensitive dependence disappears once the trajectories have converged

onto the attractor.

The existence of regions with $\sigma > 1$ for an autonomous periodic system is an important feature for the transition to chaos when the system is subjected to time-dependent forcing: this issue will be discussed in the next section. Besides, the new diagnostic method introduced in Sect. 4.2 to help analyze transition to chaos in the non-autonomous case is applied in Sect. 5.1 to the autonomous case.

## 4    The periodically forced system

For the idealized double-gyre model governed by Eq. (2), we have seen that the autonomous system given by $\varepsilon = 0$ in Eq. (3) exhibits a limit cycle when $\gamma = 1.1$; this limit cycle corresponds to the large-amplitude relaxation oscillation of Fig. 1b. On the other hand, Pierini (2014) showed that, when the same model, with the same value of $\gamma$, is subjected to periodic forcing with $\varepsilon = 0.2$ and $T_{\mathrm{p}} = 30$ yr, it exhibits chaotic, cyclostationary and cycloergodic behavior; see figures 2 and 3 therein and

the related discussion. To understand the transition to deterministically chaotic behavior induced by the forcing, we will now apply in Sect 4.1 the methodology used in Sect. 3 to the PBAs corresponding to $\gamma = 1.1$ and $T_{\mathrm{p}} = 30$ yr across the intervening parameter range $0 \leq \varepsilon \leq 0.2$. In Sect. 4.2, the transition to chaos induced by the periodic forcing will be analyzed in greater detail through an additional method developed explicitly for this purpose.

### 4.1    The pullback attractors of the forced system

Recall that the PBAs of a periodically forced system (e.g., Pierini, 2014) are always periodic, but can be either themselves periodic — in which case all trajectories are periodic and there is no mixing — or chaotic, in which case aperiodic trajectories emerge and, therefore, mixing is present. We will refer below to chaotic and nonchaotic PBAs, abbreviated as CPBAs and NCPBAs, respectively.

For the autonomous case, the time series of $P_{\Psi 3}(t)$ and the map of $\sigma(X, Y)$ — already shown in Figs. 2a and 2c, respectively

— are again plotted in Figs. 6a and 6e for the sake of comparison; Figs. 6d,h correspond to the CPBA studied by Pierini (2014). Two intermediate cases that correspond to $\epsilon = 0.05$ and $\epsilon = 0.1$ are shown in Figs. 6b,f and Figs. 6c,g, respectively. In Sect. 4.2, we will show in greater detail that the transition to chaotic behavior occurs abruptly, when crossing a critical value $\epsilon_{\mathrm{c}}$ that lies





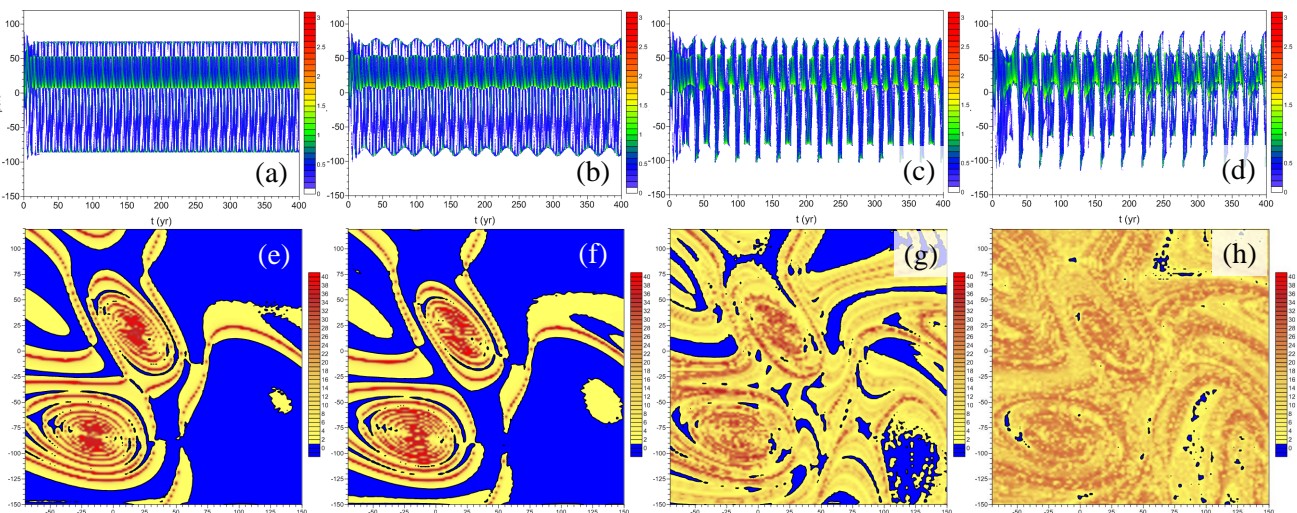

**Figure 6.** Transition from a periodic to a chaotic PBA as the amplitude $\varepsilon$ of the periodic forcing in Eq. (3) increases. (a,b,c,d) Time series of $P_{\Psi 3}$; and (e,f,g,h) maps of $\sigma$ for $\gamma = 1.1$, $T_{\mathrm{p}} = 30$ yr and $\varepsilon = 0, 0.05, 0.1$, and $0.2$, respectively.

between the two intermediate values of $0.05$ and $0.1$; here we merely provide some qualitative arguments showing that, in fact, the case $\epsilon = 0.05$ is still periodic, while the case $\epsilon = 0.1$ is chaotic.

Figure 7 provides further information on the four cases illustrated in Fig. 6. In Figs. 7a–d, the magenta dots represent the intersection with the $(\Psi_1, \Psi_3)$-plane at $t = 400$ yr of 15 000 trajectories, whose initial points (in blue) are evenly distributed in

$\Gamma$ at $t = 0$; in addition, in Figs. 7e–h, the corresponding entropy $S_{\vartheta=\Gamma}$ is plotted as a function of time, along with the number $n_{\mathrm{o}}$ of cells that are occupied by at least one point. Clearly, the structure of the PBA snapshot in Fig. 7b, for $\varepsilon = 0.05$, is very similar to that of the autonomous case in Figs. 1b and 7a, while the PBA snapshots plotted at $t = 400$ yr — for $\varepsilon = 0.1$ and $0.2$ in Figs. 7c and 7d, respectively — are quite different.

To understand better this difference, we focused in Fig. 8 on the subdomain $\vartheta_2$ of $\Gamma$ that was defined in Fig. 4c and for which

$\sigma > 1$. The autonomous case has already been analyzed in Sect. 3.1: in fact, Figs. 8a,e are equivalent to Figs. 4c,d. In the case of $\varepsilon = 0.05$, the same behavior is found, i.e., the sensitivity to initial data leads to only a compact subset of the attractor being covered; this implies the periodicity of the trajectories.

On the contrary, for $\varepsilon = 0.1$ the intersection of the trajectories at $t = 400$ yr with the $(\Psi_1, \Psi_3)$-plane, shown by the magenta dots in Fig. 8c, is virtually indistinguishable from the one that appears in Fig. 7c, when the initial data are selected in the

whole of $\Gamma$: this excellent match is an unequivocal sign of the mixing property of chaotic dynamics, as already discussed for the autonomous case $\gamma = 1.35$ in connection with Figs. 4 and 5. That the same property holds for $\varepsilon = 0.2$ in Fig. 8d is not surprising, since Pierini (2014) already recognized the model's chaotic behavior for this parameter value.

Finally, it is instructive to visualize $\Psi_3(t)$ for a couple of trajectories that are very close at $t = 0$, as done in Fig. 3 for the autonomous case. Figure 9 shows $\Psi_3(t)$ for the four cases of Figs. 6–8 and for trajectories that emerge from $\vartheta_2$. In Figs. 9a,b the





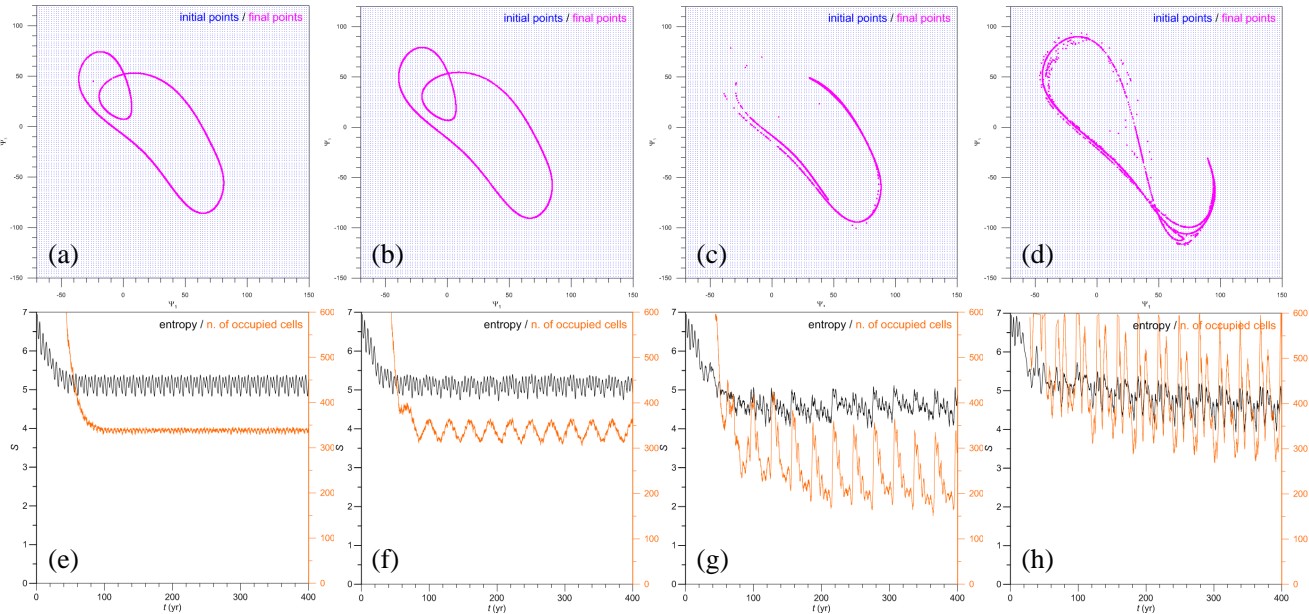

**Figure 7.** Intersection with the $(\Psi_1, \Psi_3)$-plane at $t = 400$ yr (magenta dots) of 15 000 trajectories emanating from $\Gamma$ at $t = 0$ for $\gamma = 1.1$ and $T_{\mathrm{p}} = 30$ yr. Top panels: (a) $\epsilon = 0$, (b) 0.05, (c) 0.10, and (d) 0.2. The complete set of the initial points covering $\Gamma$ is in blue. Bottom panels: (e,f,h,g) the corresponding entropy $S_{\vartheta_2}$ is plotted, along with the number $n_{\mathrm{o}}$ of occupied cells.

two trajectories are periodic, but with a phase difference. In the two chaotic cases of $\varepsilon = 0.1, 0.2$ in Figs. 9c,d, both trajectories are clearly aperiodic. In Fig. 9c, though, i.e., in the case that is closer to the transition, this aperiodicity is merely associated with a temporary shift in phase of an otherwise periodic signal within the intervals $t \simeq 40 - 120$ yr and $t \simeq 360 - 400$ yr.

The intermittent behavior seen in Fig. 9c appears — from many simulations that are not shown here — to be typical of
chaotic solutions near the transition point $\varepsilon_{\mathrm{c}}$ and suggests a possible mechanism through which chaos is induced by an external periodic forcing. For values of $\varepsilon$ just past $\varepsilon_{\mathrm{c}}$, the model still tends to behave periodically, but the external forcing is sufficiently strong to entrain a trajectory occasionally into a nearby region, where the periodicity is preserved but the phase differs by a finite amount. Since these shifts are very sensitive to the initial data, the result is a chaotic trajectory characterized by separate intervals of periodic oscillations with a different phase. This mechanism also explains why the transition to chaos leads to
a notable increase of the measure of the regions in $\Gamma$ where sensitive dependence to initial data occurs; such an increase is visually obvious when comparing Figs. 6e,f with Figs. 6g,h. As $\epsilon$ increases further, the duration of the intervals of constant phase decreases, and the oscillations tend to become more genuinely aperiodic, as seen in Fig. 9d.

This behavior is similar to the intermittency found in autonomous dissipative systems (e.g., Manneville and Pomeau, 1979; Pomeau and Manneville, 1980), in which a trajectory switches back and forth from periodic to aperiodic oscillations provided
a certain control parameter of the system — e.g., the amplitude of the steady, time-independent forcing — crosses a given





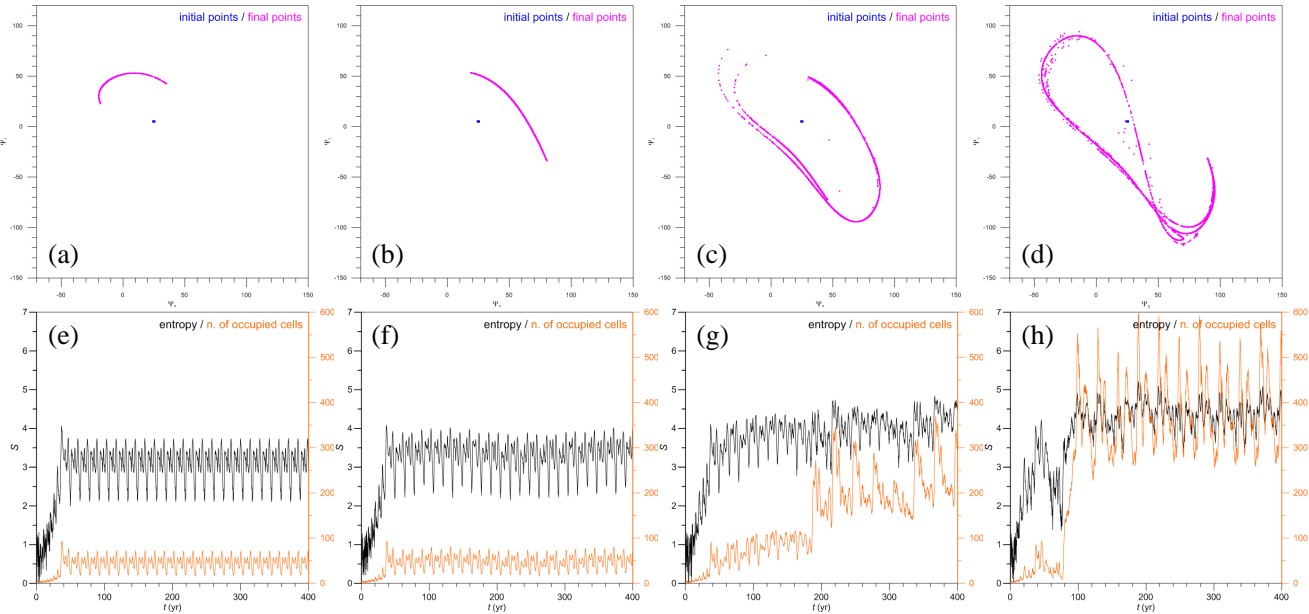

**Figure 8.** Same as Fig. 7, but for 15 000 trajectories emanating at $t = 0$ from the small rectangle $\vartheta_2$ of width $\Delta\Psi = 2$ centered at $P_2$.

threshold. In our nonautonomous system, the amplitude of the periodic forcing $\varepsilon$ plays a similar role. This transition to chaos induced by time-dependent forcing appears, therefore, to be directly linked to the existence of regions in phase space in which sensitive dependence to initial data occurs in the limit of periodic solutions. Thus, the chaotic behavior merely due to the time-dependent nature of the forcing can be traced back to the apparently paradoxical property of the autonomous system that
was emphasized in Sect. 3.2. This striking observation deserves to be analyzed in greater depth in future studies.

## 4.2 Transition to chaos studied by a cross-correlation method

Recognizing qualitatively whether a PBA is chaotic or not is relatively simple; e.g., this can be done through the heuristic arguments illustrated in Fig. 4 and Fig. 5 and through those outlined in the previous subsection and illustrated in Figs. 6–9. But how to characterize the transition from periodic to chaotic dynamics as a control parameter, such as the amplitude $\varepsilon$ of the
10 periodic forcing, changes?

We have already discussed in Sect. 3.1 the limitations of using the mean finite-time Lyapunov exponents. Here we propose a new, simple and robust method that is particularly useful in our periodic-forcing case, but can be applied also to any autonomous system and even to aperiodically forced systems; the latter situation will be addressed in Sect. 5. In Secs. 3 and 4.1, we have relied on the mixing properties of chaotic dynamics, as measured by the system's entropy $S_\vartheta$, to recognize the occurrence
of chaotic behavior. Now we rely on the emergence of aperiodic signals from a subset of $\Gamma$; this subset will necessarily be contained in the region where sensitive dependence on initial data occurs, i.e., where $\sigma > 1$.





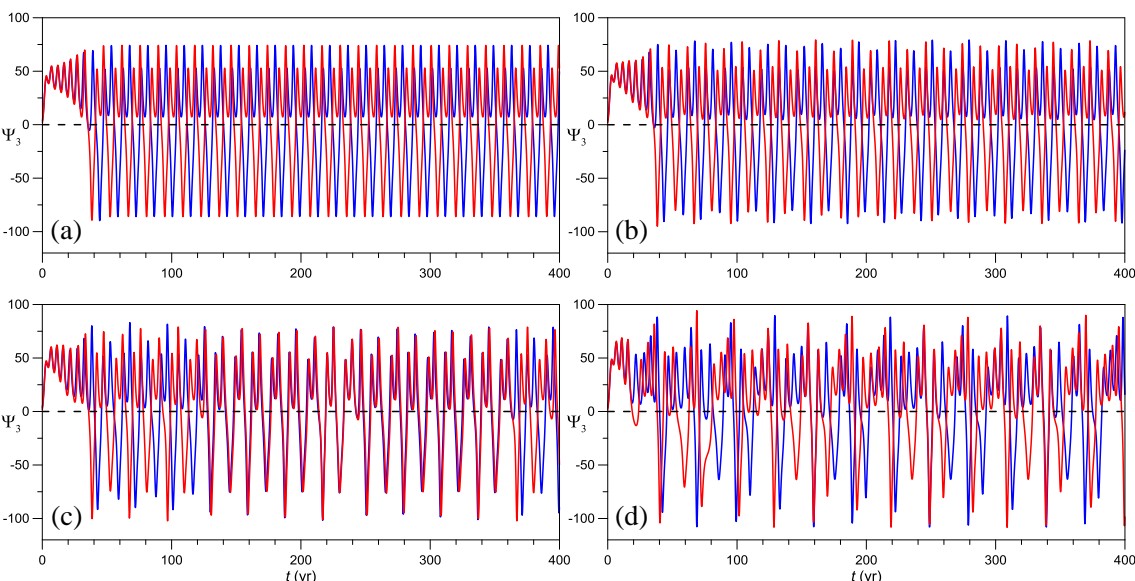

**Figure 9.** Increasing instability of trajectories as $\varepsilon$ increases. Time series of $\Psi_3(t)$ for the trajectory initialized at the point $P_2$ (red line) and at a nearby point (blue line) for $\gamma = 1.1, T_\mathrm{p} = 30$ yr, and (a–d) $\varepsilon = 0, 0.05, 0.1$, and $0.2$.

The most obvious approach would be to compute the power spectrum of each trajectory. Periodic signals can, however, be quite complex, as seen, for instance, in Figs. 9a,b; this complexity makes it quite difficult to identify a parameter whose value will distinguish, accurately and reliably, between periodic and chaotic dynamics, based solely on the Fourier spectra of a finite number of finite-length trajectories.

5      We propose a much simpler alternative method that takes advantage of the ensemble simulations carried out to obtain numerically the PBAs. Let $\bar{\Psi}_3$ and $\zeta_3$ be the mean and root-mean square values of the time series $\Psi_3(X,Y,t)$, and consider the centered and normalized anomaly time series $\widetilde{\Psi}_3(X,Y,t)$ and $\widetilde{\Psi}_3(X',Y',t)$, of $\Psi_3$, with

$$\widetilde{\Psi}_3 = \frac{\Psi_3 - \bar{\Psi}_3}{\zeta_3}, \tag{7}$$

where $(X,Y)$ and $(X',Y')$ are two points in $\Gamma$ that are near to each other, and from which these two time series emerge at
10   $t = 0$. We can then compute the cross-correlation between the two signals, after removing the initial transient, as

$$c(X,Y,\tau) = \frac{1}{T_* - 2T} \int\limits_{T}^{T_* - T} \widetilde{\Psi}_3(X,Y,t)\,\widetilde{\Psi}_3(X',Y',t+\tau)\mathrm{d}t; \tag{8}$$

in the above definition, we have dropped the dependence of $c$ on $(X',Y')$ for the sake of conciseness.





Now, if $\sigma < 1$, the two signals are periodic and virtually coincident, as seen, for instance, in Fig. 3a,c. Hence, defining the maximal cross-correlation by

$$c_{\max}(X,Y) = \max\{c(X,Y,\tau) : \tau \in [-T,T]\}, \qquad (9)$$

one will have $c_{\max} \cong 1$, with $c_{\max}$ being attained at $\tau = 0$.

5      On the other hand, if $\sigma > 1$, there are two possibilities:

- either the PBA is not chaotic, in which case all couples $(X,Y)$ and $(X',Y')$ yield two periodic and virtually equal signals, apart from a finite phase difference, as seen, for instance, in Figs. 3b and 9a,b; in this case, again, $c_{\max} \cong 1$, which will now occur at some lag $\tau \neq 0$ that depends on the phase difference;

- or the PBA is chaotic, in which case some couple yields two aperiodic and significantly different signals, as seen, for instance, in Figs. 3d and 9c,d; in this case, $c_{\max}$ will be substantially less than unity.

This alternative is illustrated in Fig. 10, with the window length $T = 50$ yr in Eq. (8), for the four couples of trajectories plotted in Figs. 9a–d, all of which were initialized in $\vartheta_2$, where $\sigma > 1$.

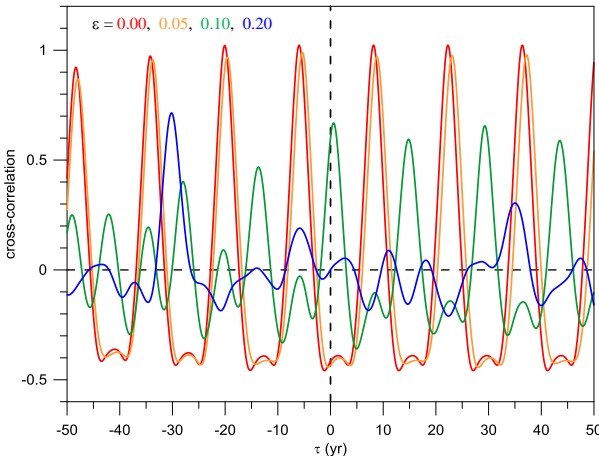

**Figure 10.** Cross-correlation $c(\tau)$ between the two initially nearby trajectories shown in Figs. 9a–d, computed for the centered and normalized anomalies $\tilde{\Psi}_3$, according to Eq. (7), for $\varepsilon = 0$ (red line), 0.05 (orange line), 0.1 (green line), and 0.2 (blue line).

It is then useful to analyze the maps of the parameter $\Theta$, defined as

$$\Theta(X,Y) = c_{\max}(X,Y). \qquad (10)$$

15    Figures 11a-d show $\Theta$ for the four cases of Figs 6–9. In the two cases that we have already identified as nonchaotic, $\Theta$ varies within a range of values $[0.95, 1.1]$ that lies very close to unity, as expected; see Figs. 11a,b. There is only a small neighborhood of $P_0 = (-16, -83)$ in which $\Theta$ is very small: this is because $P_0$ in the autonomous case is a fixed point; see Fig. 12.



The two cases that we have identified as chaotic appear here as Figs. 11c,d, and in them $\Theta$ exhibits in fact smaller values. These values lie in the range $[0.2, 0.8]$ for a large subset of the domain, where $\sigma > 1$, as shown in Figs. 6g,h. Regions in which $\sigma > 1$ but $\Theta \simeq 1$ are present as well, but the corresponding trajectories are nonetheless unstable once they have converged onto the PBA, because they will always pass sufficiently near trajectories that are chaotic, thanks to the mixing properties of the latter.

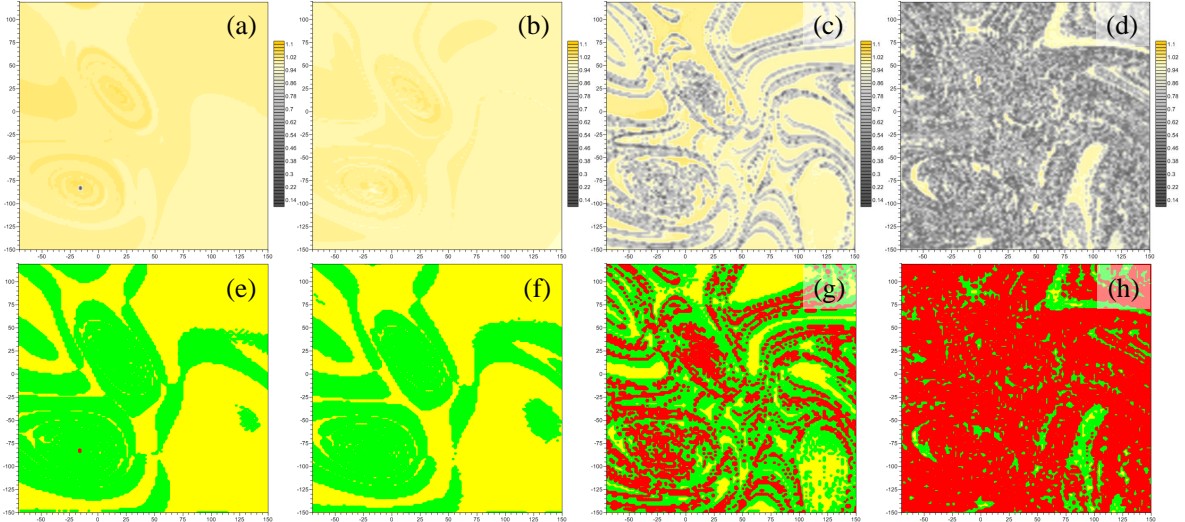

**Figure 11.** PBA diagnostics for $\gamma = 1.1$ and $T_{\mathrm{P}} = 30$ yr, with $\varepsilon = 0, 0.05, 0.1$, and $0.2$, respectively, for the two sets of four maps. Upper-row panels (a–d) show the field of $\Theta(X, Y)$, as defined in Eq. (10); the color bar for the $\Theta(X, Y)$-values is shown to the right of each panel, and it extends over the range $[0.1, 1.12]$. Lower-row panels (e–h) show the field of $\Phi(X, Y)$, as defined in Eq. (11), with the threshold value $\Theta_0 = 0.8$; here $\Phi = 1$ is colored yellow; $\Phi = 2$ is green, and $\Phi = 3$ is red. The corresponding maps of $\sigma(X, Y)$ appear in Figs. 6(e–h), respectively.

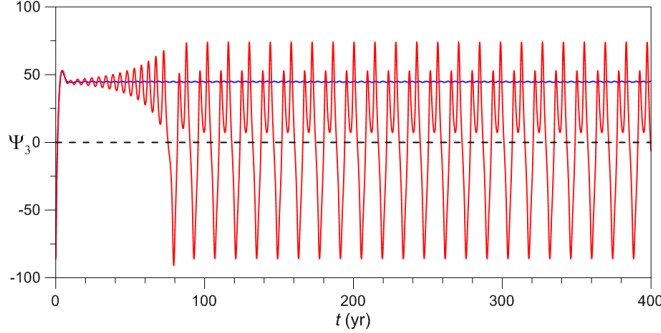

**Figure 12.** Fixed point $P_0$ of the autonomous case, with $\varepsilon = 0$. Time series of $\Psi_3$ for the trajectory initialized at $P_0 = (-16, -83)$ (blue line) and at a nearby point (solid red line) for $\gamma = 1.1$ and $T_{\mathrm{P}} = 30$ yr.



In summary, if $\Theta \sim 1$ everywhere (yellow colors) we have an NCPBA, whereas — if $\Theta$ yields values that are sufficiently smaller than unity (grey colors) — then we have a CPBA. Note that the latter case is possible only where $\sigma > 1$.

To summarize in a clear and simple way the information provided by the values of both $\sigma$ and $\Theta$, we introduce the integer-valued parameter $\Phi$, defined as follows:

$$5 \quad \Phi(X,Y) = \begin{cases} 1 & \text{if } \sigma \leqslant 1 \quad \text{(yellow)} \\ 2 & \text{if } \sigma > 1 \quad \text{and} \quad \Theta > \Theta_0 \quad \text{(green)} \\ 3 & \text{if } \sigma > 1 \quad \text{and} \quad \Theta \leqslant \Theta_0 \quad \text{(red)}. \end{cases} \tag{11}$$

Here $\Theta_0$ is a threshold value and $\Phi(X,Y)$ is plotted in Figs. 11e–h for $\Theta_0 = 0.8$.

For example, the $\sigma$-maps in Figs. 6f and 6g are fairly similar, except for the more extended warm-color region, where $\sigma > 1$, in the second map. Recall, however, that the meaning of $\sigma > 1$ is profoundly different if the system is chaotic, in which case mixing is present, as opposed to when it is not, in which case sensitive dependence to initial data is not accompanied by mixing.

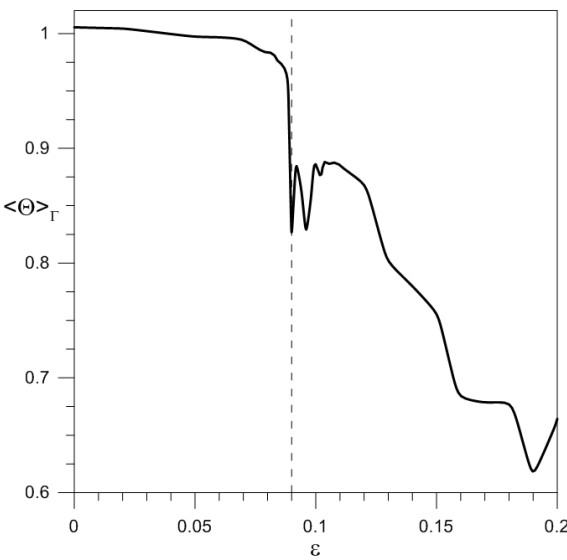

**Figure 13.** Transition from periodic to chaotic behavior, illustrated by the metric $\langle\Theta\rangle_\Gamma$ plotted vs. the amplitude $\varepsilon$ of the periodic forcing in Eq. (3); $\gamma = 1.1$ and $T_\mathrm{p} = 30$ yr.

10      This ambiguity is resolved by the use of the step function $\Phi$: if $\epsilon = 0.05$ (Fig. 11f), sensitive dependence to initial data, i.e. $\sigma > 1$, yields the value of $\Phi = 2$ (green regions), since $\Theta \simeq 1$, which tells us that the system is not chaotic. On the contrary, if $\epsilon = 0.1$ (Fig. 11g), regions with $\Phi = 2$ (in red) appear within the green regions: this implies low $\Theta$-values and therefore chaos.

We conclude by analyzing the transition from NCPBAs to CPBAs via a suitable function of the control parameter $\varepsilon$: this metric is provided by the average value $\langle\Theta\rangle_\Gamma$ of $\Theta$ over $\Gamma$. The graph of $\langle\Theta\rangle_\Gamma(\varepsilon)$ in Fig. 13 is obtained by performing many



ensemble simulations of system trajectories with many distinct values of $\varepsilon$; the latter values are chosen to lie closer to each other where the variation of $\langle\Theta\rangle_\Gamma(\varepsilon)$ is stronger. An abrupt transition from NCPBAs, with $\langle\Theta\rangle_\Gamma \simeq 1$, to CPBAs occurs at $\varepsilon_c \cong 0.09$. Many additional analyses (not shown) for values just below and above $\varepsilon_c$ confirm that this is in fact the critical value beyond which chaos sets in.

## 5 Further applications of cross-correlation diagnostics

### 5.1 Application to the autonomous system

The diagnostic method proposed in Sect. 4.2 to monitor the transition from NCPBAs to CPBAs in periodically forced systems relies on two properties: (i) in an NCPBA all trajectories are periodic; and (ii) in a CPBA aperiodic trajectories emerge from at least one subset of $\Gamma$, in which necessarily $\sigma > 1$. Thus, the same cross-correlation–based method can obviously be applied to an autonomous system as well. The method's application to the autonomous model studied in Sect. 3 will shed new light on the periodic vs. chaotic character of its solutions.

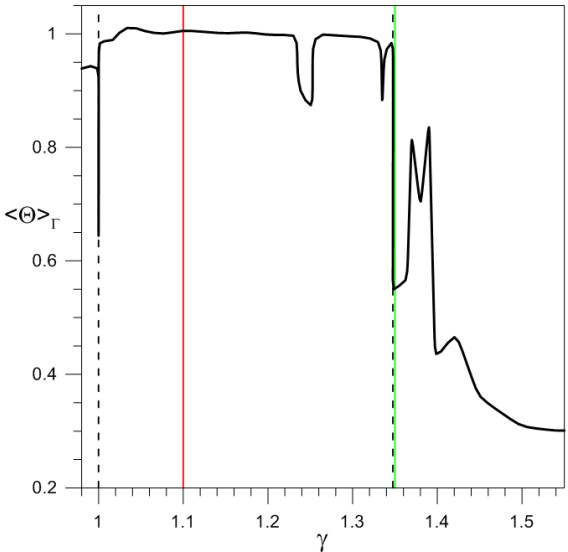

**Figure 14.** Same as Fig. 13, but for the autonomous model with $\varepsilon = 0$ and the amplitude $\gamma$ of the time-independent forcing on the abscissa. The vertical red and green lines denote the periodic and the chaotic cases, respectively, that were analyzed in Sect. 3.1; see again Fig. 1. Please see the text for the interpretation of the dashed lines corresponding to $\gamma_c = 1$ and $\gamma_0 = 1.3475$.

The graph in Fig. 14 shows $\langle\Theta\rangle_\Gamma(\gamma)$ and is obtained, like that of Fig. 13, by performing many ensemble simulations, each with a different value of $\gamma$, rather than $\varepsilon$, which equals zero in the present case. The first thing to notice is the sudden drop of $\langle\Theta\rangle$ at $\gamma = \gamma_c = 1$, where a global bifurcation separates small-amplitude limit cycles from large-amplitude relaxation





oscillations, as shown in Pierini (2011) and in Sect. 3.1 here. In addition, Pierini (2012) investigated the stochastic version of this deterministic tipping point in the case of random forcing.

The chaotic nature of the attractor for $\gamma_c$ is illustrated in Fig. 15. For $\gamma = 0.99$, the $\Psi_3$-time series exhibits the typical small-amplitude, purely periodic behavior studied in subsection 3.1, while for $\gamma = 1$ both small- and large-amplitude oscillations occur irregularly in the same time series. The behavior at $\gamma = 1.01$ illustrates the return to more regular behavior.

Thus, chaotic dynamics occurring in an extremely restricted $\gamma$-range separates two different types of limit cycles. Figure 16 shows this dramatic transition in terms of $\sigma$: the chaotic nature of the flow for $\gamma_c = 1$ is such that the warm-colored regions in which $\sigma > 1$ overwhelm the cold-colored regions, as in Fig. 2d, where $\gamma = 1.35$.

For $\gamma > 1$, the system is not chaotic — except for limited $\gamma$-intervals centered at $\gamma \simeq 1.25$ and $\gamma \simeq 1.335$ — until a new abrupt drop of $\langle \Theta \rangle_\Gamma$ at $\gamma = \gamma_0 = 1.3475$, shown by a dashed black line in Fig. 14. This drop signals the presence of chaotic attractors beyond $\gamma_0$; in particular, the chaotic case $\gamma = 1.35$, shown by the solid green line and discussed in Sect. 3, lies just after this transition. It is worth noting the large fluctuations dominate the chaotic regime.

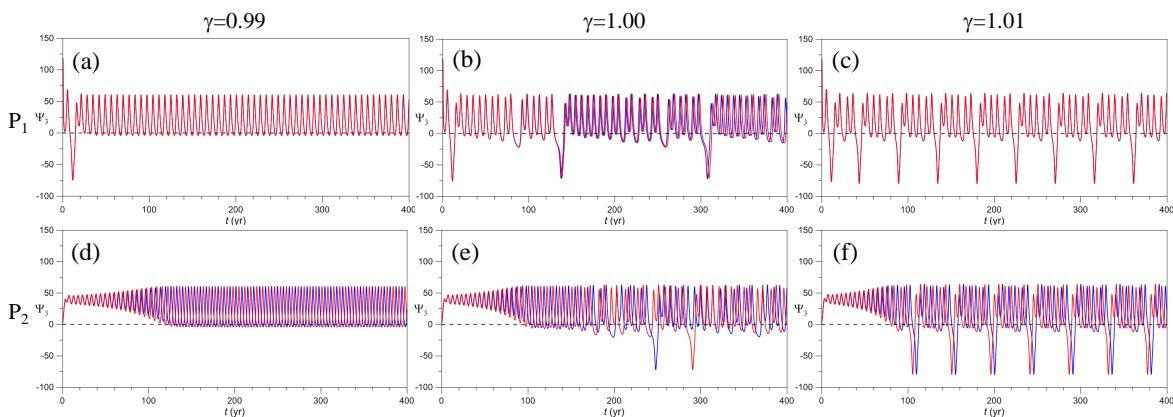

**Figure 15.** Critical transition in the autonomous system at $\gamma_c$. Panels (a,d), (b,e) and (c,f) correspond to $\gamma = 0.99, 1.0$ and $1.01$, respectively. (a,b,c) Time series of $\Psi_3$ for the trajectory initialized at $P_1$ (red line) and at a nearby point (blue line); (d,e,f) same but for trajectories initialized at $P_2$.

## 5.2 Application to an aperiodically forced system

Our cross-correlation diagnostics assumes that NCPBAs are periodic: this holds for the periodically forced case in Sect. 4.2, as well as for the autonomous case in Sect. 5.1. Thus, the method does not seem, on the face of it, to be applicable to the more realistic case of nonautonomous systems forced by an aperiodic or random process. In the present subsection, though, we will show that the method might still work by applying it to the aperiodically forced system of Pierini et al. (2016). A thorough analysis of this application is beyond the scope of the present study: we will therefore limit ourselves to analyzing the basic aspects of the problem, and leave the details for a future study.




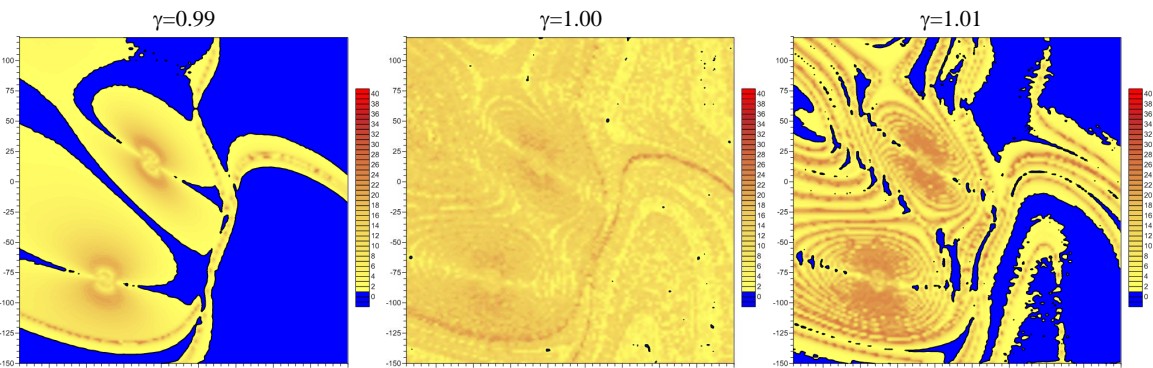

**Figure 16.** Critical transition in the autonomous system at $\gamma_c$, illustrated by maps of the mean normalized distance $\sigma$ for $\gamma = 0.99, 1.0, 1.01$.

Pierini et al. (2016) considered the same system — governed by Eq. (2), and within the same parameter regime adopted here and in Pierini (2011). The forcing, though, was aperiodic and given by:

$$G(t) = \gamma[1 + \varepsilon' f(t),] \tag{12}$$

where $\varepsilon' > 0$ is a dimensionless coefficient and $f(t)$ is a normalized, fixed realization of an Ornstein-Uhlenbeck process that

5  has been smoothed to resemble multi-annual wind-stress forcing of the mid-latitude oceans' double-gyre circulation. Figure 17 shows $G(t)$ for $\gamma = 1$ and $\varepsilon' = 0.2$.

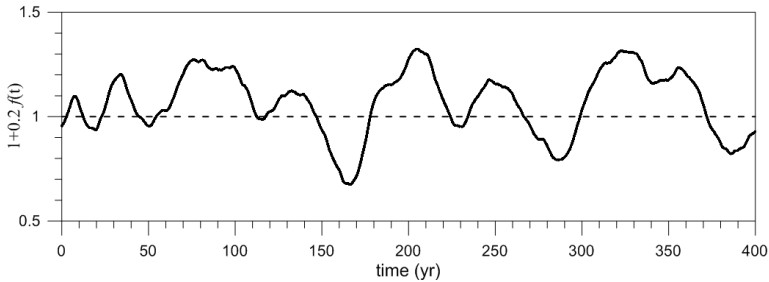

**Figure 17.** Aperiodic forcing $G(t)/\gamma$ of the idealized ocean model as defined in Eq. (12), with $\varepsilon' = 0.2$, as adopted in Pierini et al. (2016).

Figure 18 shows the evolution of two initially nearby trajectories emerging from $P_2$, along with the corresponding cross-correlation, for $\gamma = 1.1$, with $\varepsilon' = 0.05$ in the two upper panels and $\varepsilon' = 0.20$ in the two lower panels; both cases have $\sigma > 1$ and the corresponding time series of $P_{\Psi_3}$ are plotted in figures 4(h,j) of Pierini et al. (2016). The case $\varepsilon' = 0.2$ corresponds to

10  the CPBA analyzed in detail by Pierini et al. (2016). The chaotic character of the solution is clearly visible from Fig. 18c; the cross-correlation between the two signals is plotted in Fig. 18d and it is accordingly small.

On the contrary, the case $\varepsilon' = 0.05$ corresponds to an NCPBA: the two signals in Fig. 18a develop a large phase difference after the initial transient, but are virtually identical and remain coherent at all times. Now, unlike in Fig. 3b and Figs. 9a,b,




the two signals are not periodic because they are modulated by the aperiodic forcing; nevertheless, the nonchaotic character of the solution can still be highlighted by the corresponding cross-correlation in Fig. 18b, whose maximum value is $c_{\max} \simeq 1$, like in the autonomous and periodically forced case. Obviously, this is possible because the period of the modulated relaxation oscillation is much smaller than the time scale of the forcing. We conclude that the parameter $\langle \Theta \rangle_\Gamma$ can be a valuable tool for

5  monitoring the onset of chaos also in certain aperiodically forced systems.

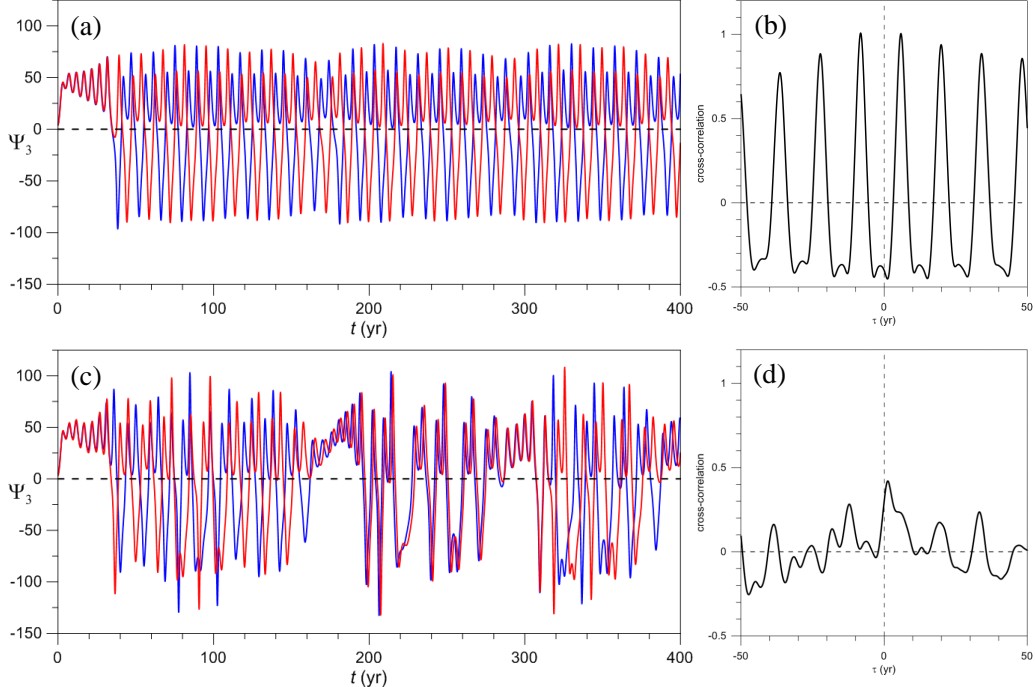

**Figure 18.** Role of the cross-correlation diagnostics in characterizing chaotic behavior for an aperiodically forced system, given by Eqs. (2) and (12); $\gamma = 1.1$. (a) Time series of $\Psi_3$ for $\varepsilon' = 0.05$ in Eq. (12); and (b) corresponding cross-correlation. (c,d) Same as (a,b) but for $\varepsilon' = 0.2$. The trajectories initialized at $P_2$ are in red and those initialized at a nearby point are in blue.

## 6 Summary and conclusions

In this paper, we studied the transition from nonchaotic to chaotic PBAs in a nonautonomous system whose autonomous limit is not chaotic. The illustrative example chosen for this general problem was a low-order quasigeostrophic model of the mid-latitude wind-driven ocean circulation, subject to periodic forcing. The model was described, and connected with previous

10  work, in Sect. 2.



We first investigated, in Sect. 3, the autonomous system, following up on the work of Pierini (2011), who obtained its bifurcation diagram. Here, ensemble simulations based on a large number of initial data and the calculation of the system's entropy allowed us to determine novel and interesting features of the system subject to steady forcing.

To do so, we used the metric $\sigma$ that was introduced by Pierini et al. (2016) and measures the time average of the distance between two trajectories that are very close at $t = 0$ on a subset $\Gamma$ of phase space. The analysis based on this metric yielded regions in $\Gamma$ with $\sigma$-values that can be either larger or less than unity. The spatial structure of these regions, cf. Figs. 2c,d here, is very similar to that of the nonautonomous case investigated by Pierini et al. (2016), as seen in Fig. 6 therein. This similarity suggests that the nonautonomous behavior of a dynamical system is profoundly influenced by the convergence properties of trajectories initialized off the attractor in the autonomous case; this finding, in turn, implies that ensemble simulations are very helpful in studying said properties.

We investigated next, still in the autonomous case, the apparent paradox of regions with $\sigma > 1$ coexisting in the periodic regime of $\gamma = 1.1$ with the expected regions of $\sigma < 1$. A large number of trajectories emanating from the small square box of Fig. 4c, for which $\sigma > 1$, evolves into the extended red line shown in the same figure at a given time, $t = 300$ yr: this line belongs to the periodic attractor shown in Fig. 1b and the entropy evolution along it oscillates periodically, cf. Fig. 4d; but it is clearly distinct from the chaotic attractor apparent as the green cloud of points in the same figure panel. Further evidence for the stability of the trajectories in this periodic case with $\sigma > 1$ is provided by Fig. 5a.

We conclude that sensitive dependence to initial data in a periodic regime may be present if the trajectories are initialized off the attractor, but that it disappears once the trajectories have converged onto the attractor. Clearly, sensitive dependence on initial data is, therewith, a necessary but not sufficient condition for chaotic behavior.

In Sect. 4, we studied the PBAs of the system subject to the periodic forcing given by Eq. (3). The PBAs were analyzed at first for four different sinusoidal-forcing amplitudes, with $\epsilon = 0, 0.05, 0.10$, and $0.2$, while using the same $\gamma = 1.1$ and $T_{\mathrm{p}} = 30$ yr.

We found that the first two cases, namely $\epsilon = 0$ and $0.05$, are not chaotic, while the other two, namely $\epsilon = 0.10$ and $0.20$, are chaotic: A large number of trajectories emanating from the small square box where $\sigma > 1$ in Figs. 8a–d evolves — depending on the value of $\varepsilon$ — into two very different fixed-instant subsets of the model's PBA, with the snapshot taken at $t = 400$ yr, i.e. after convergence of the trajectories to the PBA. In the first two cases, this snapshot is a curved-line segment that belongs to the PBA, while in the latter two cases it covers the whole PBA, due to the typical mixing property of chaos.

An analysis of the trajectories, as shown in Fig. 9, for instance, indicates that the transition to chaos occurs via an intermittent emergence of periodic oscillations with different phases; see again Fig. 9c. We have shown that, for values of $\epsilon$ in the chaotic regime just above the transition, the periodic character of the system is still predominant, but the external forcing is now sufficiently strong to cause a trajectory's occasionally shifting to a phase space region in which a different phase prevails: since the shifts are very sensitive to the initial data, the result is a chaotic trajectory characterized by irregular jumps of the oscillatory solutions between distinct phases.

In Sect. 4.2, we introduced a novel diagnostic method for the study of the transition between nonchaotic and chaotic behavior as the amplitude $\epsilon$ of the periodic forcing increases. The method's basic idea is that in a nonchaotic regime all trajectories are periodic, while in a chaotic regime aperiodic trajectories emerge from at least some subset of $\Gamma$, in which necessarily $\sigma > 1$.





Hence, a systematic recognition of the periodic or aperiodic character of all the trajectories allows one to diagnose which of these two types of behavior occurs or whether the two actually coexist.

A simple and robust way to do this is to compute the cross-correlation $c(X, Y, X', Y', \tau)$ at lag $\tau$ between two initially nearby trajectories started at $(X, Y)$ and $(X', Y')$ in $\Gamma$, then compute its maximum $c_{\max}(X, Y, X', Y') = \Theta(X, Y)$ over $\tau$. If $\Theta(X, Y) \simeq 1$ everywhere in $\Gamma$, then the system is periodic; on the contrary, if $\Theta$ is appreciably smaller than unity in some subset of $\Gamma$, the system is chaotic. The diagram of the average $\langle \Theta \rangle_\Gamma (\epsilon)$ of $\Theta(X, Y)$ over $\Gamma$, as plotted in Fig. 13, reveals an abrupt transition to chaos at $\epsilon_c \simeq 0.09$.

This cross-correlation–based method has also been applied to the autonomous system in Sect. 5.1 and to the aperiodically forced system studied by Pierini et al. (2016) since, in fact, the conditions under which the method applies are satisfied in these two cases as well. For the autonomous system with steady forcing, the diagram of $\langle \Theta \rangle_\Gamma (\gamma)$ in Fig. 14 reveals that chaotic dynamics occurs at first within an extremely restricted range centered at the global bifurcation point $\gamma = \gamma_c = 1$, which separates small-amplitude, fairly smooth oscillations below $\gamma_c$ from large-amplitude relaxation oscillations above it. A considerably broader range of chaotic behavior occurs in the autonomous case for values of $\gamma$ greater than a threshold $\gamma_0 = 1.3475$.

In the aperiodic case, the cross-correlation method seems to apply at least when the system's intrinsic periodicity and the characteristic time scale of the external forcing are sufficiently well separated from each other, which is the case investigated by Pierini et al. (2016), cf. Fig. 17 here. Once more, the cross-correlations in Figs. 18c,d agree remarkably well with the character of the model trajectories in Figs. 18a,b.

Finally, the coexistence of local PBAs with chaotic vs. nonchaotic behavior within a global PBA — as first described by Pierini et al. (2016) in the aperiodic-forcing case — was confirmed here for the periodically forced case, cf. Figs. 6 and 11. This situation was explored in greater depth in App. A for an even simpler, weakly dissipative nonlinear model, namely a Van der Pol–Duffing oscillator (e.g., Jackson, 1991), and additional references were given for this type of PBA bistability.

Overall, this paper provides additional insights into the complex and varied behavior that arises even in highly idealized atmospheric, oceanic and climate models from the interaction of nonlinear intrinsic dynamics with various types of external forcing. In addition, it stresses the importance of using the framework of nonautonomous dynamical systems and of their PBAs for a deeper understanding of this complexity and variety.

*Acknowledgements.* SP would like to thank the "Università di Napoli Parthenope" for having supported his visit to the University of California at Los Angeles in August 2017 through grants D.R. 539 (29-6-2016) and D.R. 953 (28-11-2016, DSTE315). This work has been partially supported by the Office of Naval Research (ONR) Multidisciplinary University Research Initiative (MURI) grants N00014-12-1-0911 and N00014-16-1-2073 (MDC and MG). MDC also also gratefully acknowledges support from the National Science Foundation (NSF) grants OCE-1243175, OCE-1658357 and DMS-1616981.



## Appendix A: Coexistence of pullback attractors in a Van der Pol-Duffing oscillator

The purpose of this appendix is to provide further insight into the coexistence of local PBAs with quite different stability properties, as illustrated in the main text by Figs. 6 and 11. In complex autonomous systems, several local attractors may coexist for a given set of the system's parameters; each of these attractors possesses attracting sets of initial data that are

typically separated by fractal boundaries (Grebogi et al., 1987). Basins of attraction with fractal boundaries have consequences for predictability: uncertainties on the initial state $x_0$ may result in different types of dynamical behavior, depending on which basin $x_0$ lies in; see, for instance, Roques and Chekroun (2011). When the number of the coexisting attractors is two, one speaks of bistability, and multistability refers colloquially to more than two coexisting attractors.

Our goal here is to illustrate how multistability for nonautonomous systems manifests itself unambiguously through the

existence of disjoint local PBAs. In the case of periodically forced systems, such as considered in this article, similar results can be inferred, of course, from the analysis of Poincaré maps. Nevertheless, the presence of other frequencies in the internal dynamics, the external forcing or the noise may render the analysis of Poincaré maps difficult, whereas the framework of PBAs includes naturally such additional levels of complexity (Chekroun et al., 2018). Pierini et al. (2016), for instance, demonstrated already the usefulness of this framework in the context of bistability.

Furthermore, the purpose of this Appendix is also to illustrate that multistability of local PBAs arises not only for our quasi-geostrophic model, as discussed in the paper's main text. More generally, PBA coexistence occurs fairly often for externally forced systems, albeit a careful analysis of the flow's dependence on initial data may be required in practice in order to conclude on multistability.

A paradigm of multistability is provided by dissipative nonlinear systems that become Hamiltonian in the limit of vanishing

dissipation, as is the case, for instance, in celestial mechanics (e.g., Ghil and Wolansky, 1992). In this situation, it is expected that the number of coexisting attractors exceeds any fixed bound in approaching this limit, as documented for various nearly-integrable maps and flows, cf. (Feudel et al., 1996; Zaslavsky and Edelman, 2008; Celletti, 2009; Pisarchik and Feudel, 2014; Dudkowski et al., 2016) and references therein.

To illustrate this multistability phenomenon, we consider the following periodically forced Van der Pol–Duffing oscillator,

given by the second-order nonlinear ODE

$$\ddot{x} = \mu(1 - \dot{x}^2)x - bx^3 + F\sin(\omega t), \tag{A1}$$

where $\mu$, $b$, $F$ and $\omega$ are positive constants that determine the dynamical behavior of the the system. This system is non-autonomous and its PBAs are analyzed hereafter in the $(x, \dot{x})$-plane. This nonlinear ODE arises in various applications such as in engineering, electronics, biology, and neurology (Jackson, 1991; Kozlov et al., 1999; Kuznetsov et al., 2009). It combines

the nonlinearity of the dissipation $-\mu\dot{x}^2 x$, which characterizes the Van der Pol (1920) oscillator with that of the internal force $-bx^3$, which characterizes the Duffing (1918) oscillator.

Multistability was already numerically documented for Eq. (A1) by relying on Poincaré maps (Venkatesan and Lakshmanan, 1997; Dudkowski et al., 2016). In our calculations, we have followed (Dudkowski et al., 2016) and assumed $\mu = 0.2$, $F = b =$



1.0, and $\omega = 0.955$. While this parameter regime does not correspond to the limit of vanishing dissipation that was mentioned above, it still allows for a coexistence of PBAs, given a careful choice of initial states.

The numerical protocol followed to analyze multistability for Eq. (A1) in terms of PBAs is described next. First, the initial data have been drawn uniformly in the two disjoint domains $D_1$ and $D_2$ of the $(x, \dot{x})$-plane, with

$$D_1 = [-2.5, -1.5] \times [-2.5, -1.5], D_2 = [-0.34, -0.33] \times [-0.34, -0.33]. \tag{A2}$$

6 000 initial data from each domain were propagated according to Eq. (A1). The ODE was integrated using a Runge-Kutta fourth-order method with a constant time step $\Delta t = 10^{-2}$, generating a total of 12 000 trajectories, and keeping $10^6$ data points for each, after removal of the transient.

The majority of the initial data taken in the smaller domain $D_2$ leads to a quasiperiodic orbit, while each of the 6 000 initial data taken in $D_1$ leads to a chaotic trajectory — as do a few "rare" initial data from $D_2$. An example of such a quasiperiodic trajectory is shown in blue in Fig. A1(a), within the $(x, \dot{x})$ phase plane. This blue trajectory is superimposed upon a red, chaotic trajectory emanating from an initial point taken in $D_1$. The corresponding power spectra are shown in Fig. A1(b), with the same blue and red color coding. The chaotic trajectory is clearly more diffuse within the phase plane than its quasiperiodic counterpart, and its power spectrum is quite a bit noisier. Both of these features are well known to be symptomatic of deterministic chaos (Eckmann and Ruelle, 1985).

By allowing the quasiperiodic trajectories that emanate from $D_2$ to evolve up to $t = 2\,000$, one obtains the set of blue points shown in Fig. A2. Somewhat surprisingly, this set does not form a closed curve: it is actually the union of quasiperiodic orbits that — while sharing the same fundamental combinations of frequencies shown in Fig. A1(b) (for the blue curve) — may exhibit a power spectrum that contains a multitude of local variations reminiscent of those exhibited by power spectra of the chaotic orbits, i.e. similar to those shown by the red curve in Fig. A1(b). Such quasiperiodic orbits are wrapped onto a torus with a larger minor radius that results typically in broader strips in the phase plane (not shown) than those displayed by the blue orbit of Fig. A1(a).

The initial data taken in $D_1$, when allowed to flow according to Eq. (A1), lead to a totally different local PBA that is formed by the red points shown in Fig. A2. Although the approximation of this local PBA shown here is relatively sparse, one can clearly discern the fact that its constitutive points are arranged according to a stretching and folding pattern that is typical of nonlinear, chaotic dynamics in the autonomous, as well as in the nonautonomous setting (Chekroun et al., 2018).

The features that we find here to be exhibited by the local chaotic PBA are highly reminiscent of those that were obtained, in this deterministic case, by applying a standard Poincaré section analysis, cf. (Dudkowski et al., 2016, Fig. 17(c)). In the presence of noise, though, the fine structure of the PBAs that results from stretching and folding in phase space, is still captured by the PBA framework (Chekroun et al., 2018), whereas a Poincaré-map approach would lead only to a cloud of points with no particular geometric structure. This statement was numerically illustrated in (Chekroun et al., 2011) by contrasting, in their Fig. 7, the upper-right panel vs. the six lower panels of the figure.




Finally, we emphasize that it is not the disjointness of the two domains, $D_1$ and $D_2$, that leads to the two distinct types of PBA, chaotic and quasiperiodic.[2] Indeed, as mentioned earlier, even though the area of $D_2$ is small, it still contains initial data whose evolution lands within the local PBA associated with chaos, i.e., with the other red points shown in Fig. A2.

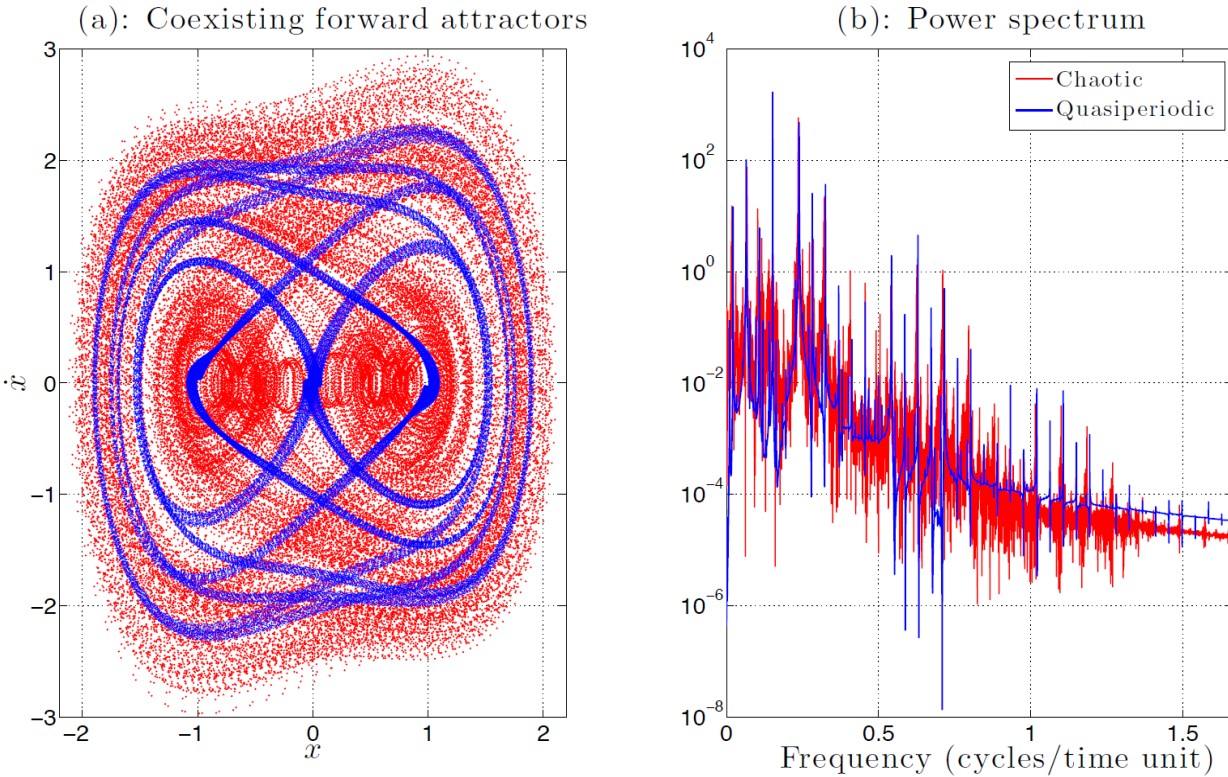

**Figure A1.** Coexistence of local forward attractors. (a) Quasiperiodic forward attractor (blue) and chaotic attractor (red). (b) Power spectrum associated with the quasiperiodic orbit (blue) and the chaotic one (red).

*Competing interests.* The authors declare no conflict of interest.

---

[2]Other such quasiperiodic PBAs exist (not shown), whereas only one chaotic PBA seems to exist for the parameter regime analyzed herein.





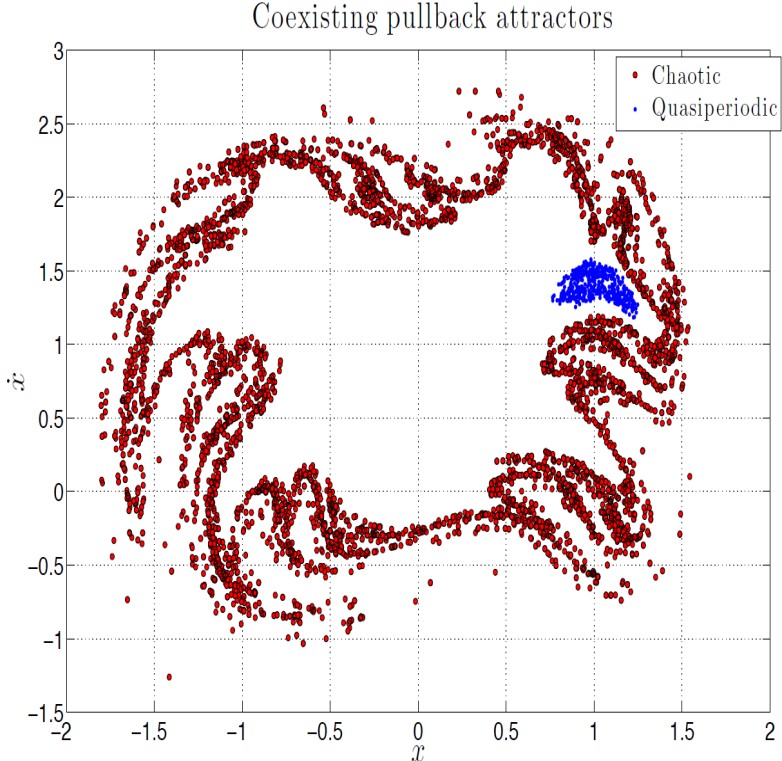

**Figure A2.** Coexistence of local PBAs. The initial data leading to quasiperiodic (resp. chaotic) orbits are taken from the small domain $D_2$ (resp. $D_1$) described by Eq. (A2). The snapshot of the PBAs shown here is taken at the fixed time $t = 2\,000$.

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
