# Peer review of "The onset of chaos in nonautonomous dissipative dynamical systems: A low-order ocean–model case study\"

_Nonlinear Processes in Geophysics, 2018_

## Referee Comment (RC1) · Anonymous Referee #1 · 29 May 2018

The authors investigate the onset of chaos in general nonautonomous dissipative systems by means of an ensemble approach, on the example of a low-order ocean model. They use several indicators, among which the cross-correlation between two nearby trajectories is particularly promising since there is no consensus on how to identify chaos in systems of arbitrary time-dependence. The paper is interesting, and well-written. It certainly deserves publication after the clarification of a few points,

The classical literature on dynamical systems (see e.g. Eckmann-Ruelle or Ott) clearly considers chaotic attractors of periodically driven systems usual, i.e. non-pullback, attractors, since they can be observed on stroboscopic maps taken with the period of

driving on which the dynamics appears to be autonomous. With the exception of a short section (5.2), all the examples investigated in the paper are autonomous or periodically driven, their attractos are thus non-pullback attractors. The pullback concept is meaningful to reserve for systems of arbirtrary, non-periodic time-dependence. The authors represent, however, the attractors by ensembles instead of long time series, although the two methods are known to be equivalent in cases with constant or time-periodic driving. The novelty of their approach is that they do not take advantage of the simply form of the time-dependce, their conclusions might thus be valid for systems of arbitrary time dependence, too. I recommend to make clear in the text that the authors investigate non-pullback attractors as if they were pullback.

Further remarks p1, line 18: not all autonomous dissipative system exhibit period doubling cascade, etc. Think of nondriven mechanical systems in the presence of friction: all motion stops, there is no asympotic dynamics. The sentence should be reformulated.

Fig.1: what is the time t at which the attractors are plotted?

Eq.(5): This quantity is similar to the broadly used finite time Lyapunov exponent. Why do you think that sigma is better suited here?

Fig.2: what is the value of T*? How do the plots change if T* changes? Why exactly this T* is taken?

Fig.3, panel c: this is a chaotic case, and the blue curve cannot be distinguished from the red? This contradicts Fig.4a where the green points are spread even if started from the vicinity of P_1.

p7, l4: Fig.4a is taken at t=400, but the caption says t=300. What is the correct statement?

Fig.4a: I do not see the red dots mentioned in line 5 of page 7.

Fig.4b,d: I wonder if the use of this entropy is useful if it gives different values for the

same chaotic attractor even after 400 time units. We know that S should asymptotically converge to a costant (exactly since the driving is constant). What if we take more initial points in the small square used in Fig.4a?

p.8, lines 2-5: you speak about a "sensitive dependence on initial data". I agree, there is some dependence but this is certainly different from the traditional sensitivity since the letter holds on the chaotic attractors. Yours is there for a periodic motion and disappears, as you say in line 5, on the attractor. It might be useful to use a different terminology here.

p9, line17: what is the meaning of "spans the attractor roughly six times"?

Fig.10: What is T and T* here? How do the results depend on their choice?

Section 5.2: The system treated here is governed by a real pullback attractor. The time-dependene shown in Fig. 17. is a general one but does not possess any drift. Since systems with drift are important in undertsanding e.g. climate change, the question arises: would <theta>_Gamma be a useful indicator also for this type of systems?

p21, line 7: "In this paper, we studied the transition from nonchaotic to chaotic PBAs in a nonautonomous system whose autonomous limit is is not chaotic. I am lost: is not the attractor of Fig. 1c chaotic?

p25,line 17: The sentence "the union of quasiperiodic orbits that . . . may exhibit a power spectrum that contains a multitude of local variations reminescent of those exhibited by power spectra of the chaotic orbit" is not convincing. Is the attractor quasiperiodic or chaotic? The blue dots in Fig. A2 do not clarify the situation either. Therfore, here at least I recommend to show this attractor also as the result of a long time series (after an appropriate removel of transients).

---

## Referee Comment (RC2) · Anonymous Referee #2 · 1 Jun 2018

In this paper, a new method that characterizes the transition from periodic to chaotic dynamics in non-autonomous dynamical systems is presented. The paper is generally well written and can be published with minor revisions. Section 3.2 is not needed in my opinion. I wander if it is correct to interpret a transient growth as a "sensitive dependence on initial conditions".

Pg 5, line 4: please use the term " time evolution" instead of "time series", as in Pierini 2016, throughout the text.

Pg 5 line 7: (X,Y) is not defined. Is it the initial condition of ($\Psi 1, \Psi 3$) as in eq. 5?

Pg 6 line 5: why $\sigma > 1$ initially? $\sigma = 1$ initially by definition (5).

[Figure]

Pg 7, Figure 3c. with $\gamma$=1.35 you get chaos. How is it possible that two initially close trajectories never diverge? This contradicts the statement at the beginning of pg 9.

Pg 9, line 7 and following lines. I understand that you cannot compute with enough accuracy the first Lyapunov exponent (why? I found it strange for such a low dimensional system). If so, you cannot say that "the results prove unequivocally the assumption.."

Pg 9 line 11: illustrated instead of summarized.

Pg 9 caption of figure 5 and in other parts of the text: "non chaotic".

Pg 9 line 17: 4T$\Delta$=100

Pg 10 line 25. It is a contradictory statement. Please explain it better.

Pg 15 eq 10: I do not see the reason to introduce a new symbol for Cmax. There are already a lot of symbols to keep in mind.

Pg18, last line: <$\theta$> → <$\theta$>$\Gamma$

---

## Author Comment (AC1) · 15 Jul 2018

We would like to thank the reviewer for his/her valuable comments that have helped us improve the manuscript. Please find below our responses to the reviewer's comments.

———————————————————————-

REVIEWER: p1, line 18: not all autonomous dissipative system exhibit period doubling cascade, etc. Think of nondriven mechanical systems in the presence of friction: all motion stops, there is no asymptotic dynamics. The sentence should be reformulated.

[Figure]

RESPONSE: We thank the reviewer for this comment. We have added "in the presence of time-independent forcing", now on p. 1, l. 20 of the revised ms. " [. . .] in autonomous dissipative systems - in the presence of time-independent forcing - include [. . .]".

————————————————————————-

REVIEWER: Fig.1: what is the time t at which the attractors are plotted?

RESPONSE: The attractors are time-independent, but of course only after spinup. Following this useful comment we have now specified the time in the text (p. 5, l. 2): "...and it is plotted at t = T* = 400 yr, i.e. after spinup."

————————————————————————-

REVIEWER: Eq.(5): This quantity is similar to the broadly used finite time Lyapunov exponent. Why do you think that sigma is better suited here?

RESPONSE: The metric sigma provides information that is different from that provided by the Lyapunov exponents. Why have we used sigma here? We had pointed out in this paper's original version - and still do so - that "Pierini et al. (2016) found the quantity sigma to be a good indicator of the degree of sensitivity of the system's evolution with respect to the initial state during the phase of convergence to the attractor." In addition, following this useful comment, we have now stressed, furthermore, that (p. 6, l. 1-4): "The determination of the PBAs of the periodically forced system and the application of the new qualitative and quantitative diagnostic methods proposed in section 4 need an analysis of the behavior of trajectories that lie at t = 0 on a given subset Omega of phase space, as is the case when calculating sigma(X,Y) above. Thus, investigating the behavior of model trajectories as they emerge from Omega is the most unifying and distinctive feature of the present model study."

————————————————————————-

REVIEWER: Fig.2: what is the value of T*? How do the plots change if T* changes? Why exactly this T* is taken?

RESPONSE: The value of T* (400 yr) was, and still is, defined in section 2, between Eqs. (3) and (4). The plots of Fig. 2 are independent of T* provided the latter it is sufficiently greater than the spinup time, but as clearly shown by the graphs of Fig. 3, this is definitely achieved for our choice. Following this useful reviewer's comment, this is now specified on p. 4, l. 11: "... as shown in Fig. 3 and 9 below, T* is much greater than the spinup time in all cases."

————————————————————————-

REVIEWER: Fig.3, panel c: this is a chaotic case, and the blue curve cannot be distinguished from the red?

RESPONSE: Yes, the red and blue curves are virtually coincident; this is what happens for trajectories leaving from the very restricted cold-color regions of Fig. 2d; on the other hand, this possibility is not surprising in view of the results of Pierini et al. (2016) (p.7, l. 3): "If sigma < 1, as is the case near P1, the two trajectories are virtually coincident (Fig. 3c)." Of course, these trajectories are nonetheless unstable once they have converged onto the PBA.

REVIEWER: This contradicts Fig.4a where the green points are spread even if started from the vicinity of P_1.

RESPONSE: We thank the reviewer for this comment. For the response please see p. 8, l. 3-6: "In the chaotic case gamma=1.35, sigma<=1 for 43% of the points contained in theta1, while sigma> 1 for the remaining points. The evolution of the former lead to the localized blue dots in Figure 4a while the evolution of the latter lead to the green dots scattered over the strange attractor. The green line of Figure 4b, giving Stheta1 computed with all the trajectories, shows the gradual spreading of the initial points with sigma> 1." Please see also p. 9, l. 7-10 (see a further response below).

————————————————————————-

REVIEWER: p7, l4: Fig.4a is taken at t=400, but the caption says t=300. What is the

correct statement?

RESPONSE: The correct statement in both cases is t = 300 y. Thank you.

——————————————————————————-

REVIEWER: Fig.4a: I do not see the red dots mentioned in line 5 of page 7.

RESPONSE: As explained in the discussion of Fig. 4a, "...The entropy of the periodic case gamma=1.1, characterized initially by sigma<1, oscillates between 0 and 1, with the final evolution limited to virtually a single cell over the limit cycle; ... ". You can recognize the small cluster at (Psi1,Psi2)ïĄĂ(7,12), but we understand that this is not easy to be noticed. So, following this useful comment, now we have added a circle in the new Fig. 4a and have mentioned it in the text (p. 8, l. 2): "... with the final evolution limited to virtually a single cell over the limit cycle; the latter cell is enclosed in the red circle of Figure 4a."

——————————————————————————-

REVIEWER: Fig.4b,d: I wonder if the use of this entropy is useful if it gives different values for the same chaotic attractor even after 400 time units. We know that S should asymptotically converge to a costant (exactly since the driving is constant). What if we take more initial points in the small square used in Fig.4a?

RESPONSE: We thank the reviewer for this comment. For our response please read p. 9, l. 7-10: "Finally, it is worth stressing that, since the forcing is constant, the range of variability of the entropy in the chaotic case with sigma>1 must tend to zero as the number of points tends to infinity. This tendency is clearly illustrated by the green line of Figure 4d. On the other hand, the range of variability of Stheta1 in the chaotic case (Figure 4b) is still quite large after 400 yr because, as pointed out above, the number of points with sigma>1 contained in theta1 is relatively small."

——————————————————————————-

REVIEWER: p.8, lines 2-5: you speak about a "sensitive dependence on initial data". I agree, there is some dependence but this is certainly different from the traditional sensitivity since the letter holds on the chaotic attractors. Yours is there for a periodic motion and disappears, as you say in line 5, on the attractor. It might be useful to use a different terminology here.

RESPONSE: We thank the reviewer for this comment. Now we have used the expression "sensitive phase dependence on initial data" when referring to the non-chaotic case with sigma>1. Besides, we have also modified the first paragraph of section 3.2 in order to better characterize this specific form of sensitivity (p. 9, l. 12-20): "We conclude the analysis of the autonomous system by discussing an apparent paradox. We have just seen that, in regions of Gamma where sigma> 1, the trajectories for gamma = 1.1 exhibit sensitive phase dependence on initial data, as shown, for instance, by Fig. 3b, by the red dots in Fig. 4c and by the red curve in Fig. 4d. Sensitive dependence on initial data is usually associated with chaotic dynamics, but in this case the dynamics is periodic. This paradox is resolved by noting that such sensitivity concerns only the phase of the periodic trajectories, as already noticed in the previous subsection and, in addition, it occurs only if the initial data lie outside the attractor, e.g, elsewhere on Gamma; on the attractor, this phase sensitivity disappears, as we will show below. On the contrary, in the chaotic case gamma = 1.35, the sensitivity to initial data for trajectories with sigma > 1 always holds, off the attractor as well as on it, in excellent agreement with the chaotic character of the dynamics in the latter case."

In any case, in the same section we have clearly explained the fundamental difference between this sensitivity and that associated with chaotic dynamics, so we believe there can be no misunderstanding.

————————————————————————————-

REVIEWER: p9, line17: what is the meaning of "spans the attractor roughly six times"?

RESPONSE: We have now improved the explanation (p. 10, l. 2-3): "... indeed, these

points that start from t = 300 yr, evolve anticlockwise around the attractor, covering it roughly six times during the interval 4TDelta = 100 yr that separates the first snapshot from the last one."

————————————————————————-

REVIEWER: Fig.10: What is T and T* here? How do the results depend on their choice?

RESPONSE: Please see p. 15, l. 16-17 for the response to this comment: "... here T* = 400 yr is again the maximum integration time, and −T<=tau<=T, while T = 50 yr; once more, the following results are independent of T, provided it is sufficiently larger than the typical time scale of the phenomenon."

————————————————————————-

REVIEWER: Section 5.2: The system treated here is governed by a real pullback attractor. The time-dependene shown in Fig. 17. is a general one but does not possess any drift. Since systems with drift are important in undertsanding e.g. climate change, the question arises: would <theta>_Gamma be a useful indicator also for this type of systems?

RESPONSE: We thank the reviewer for this comment. First of all we have more clearly stated that the new diagnostic method proposed in this study can be applied to a wide class of aperiodically forced systems. Moreover, we have now stressed that the method can find application to the specific, and very interesting case of systems possessing a drift (p. 21, l. 20-21): "For example, this diagnostic method can be applied to study the onset of chaos in systems that possess a drift mimicking global warming and other climate change scenarios (as done, for instance, in Drótos et al., 2015)."

————————————————————————-

REVIEWER: p21, line 7: "In this paper, we studied the transition from nonchaotic to chaotic PBAs in a nonautonomous system whose autonomous limit is is not chaotic. I

am lost: is not the attractor of Fig. 1c chaotic?

RESPONSE: But in fact our analysis of section 4 refers to the case gamma=1.1, whose attractor is that of Fig. 1b. This comment is very useful, however, because, to avoid confusion, we followed up on it by specifying that "... chaos is induced by the periodic forcing" (p. 22, l. 3).

————————————————————–

REVIEWER: p25,line 17: The sentence "the union of quasiperiodic orbits that . . . may exhibit a power spectrum that contains a multitude of local variations reminescent of those exhibited by power spectra of the chaotic orbit" is not convincing. Is the attractor quasiperiodic or chaotic? The blue dots in Fig. A2 do not clarify the situation either. Therfore, here at least I recommend to show this attractor also as the result of a long time series (after an appropriate removel of transients).

RESPONSE: This paragraph was not sufficiently clear and it has been revised. We thank the reviewer for pointing out the need for further clarification. In particular, Fig. A3 has been added; this figure illustrates better, and in a more standard way, the distinction between quasi-periodic and chaotic orbits. Please see the new discussion on p. 26, l. 11-24: "By allowing the quasiperiodic trajectories that emanate from D2 to evolve up to t = 2000, one obtains the set of blue points shown in Fig. A2. Somewhat surprisingly, this set does not form a closed curve: each blue dot in Fig. A2 corresponds actually to the state at t = 2000 in the phase plane of a quasi-periodic orbit. One such orbit is represented in blue in Fig. A1(a), after removal of the transient dynamics. Each blue dot in Fig. A2 corresponds to a different quasiperiodic orbit, whose frequency characteristics may slightly change from one blue dot to another. All these quasiperiodic orbits share, however, a spectral signature that resembles the one shown by the blue curve in Fig. A1(b). To illustrate further the distinction between quasiperiodic and chaotic orbits, the return maps for the minima of the x(t)-variable have been computed. As is well-known (e.g., Strogatz, 2018), if the return map contains just one

point, the solution is periodic in time, with all minima having the exact same value, and the period of the oscillation can be estimated by calculating the time interval between two consecutive minima. If the return map contains continuous-looking curves that fill up with more and more points as the length of the orbit increases, the solution is quasi-periodic, while the presence of folds and self-similarity in the return map provides strong evidence for chaotic solutions. For the blue and red trajectories of Fig. A1(a), we plot the corresponding return maps in Figs. A3(a) and (b), respectively. The two plots clearly discriminate between the quasiperiodic nature of the former and the chaotic one of the latter solution."

———————————————————————————————-
————————————————————————

**Fig. 1.** Figure 4 (modified)

(a): Return map of the minima

(b): Return map of the minima

**Fig. 2.** Figure A3 (new)

---

## Author Comment (AC2) · 15 Jul 2018

We would like to thank the reviewer for his/her valuable comments that have helped us improve the manuscript. Please find below our responses to the reviewer's comments.

————————————————————————-

REVIEWER: Section 3.2 is not needed in my opinion. I wander if it is correct to interpret a transient growth as a "sensitive dependence on initial conditions".

RESPONSE: We thank the reviewer for this comment. We have used instead the

expression "sensitive phase dependence on initial data" when referring to the non-chaotic case with sigma>1. Besides, we have also modified the first paragraph of section 3.2 in order to better characterize this specific form of sensitivity (p. 9, l. 12-20): "We conclude the analysis of the autonomous system by discussing an apparent paradox. We have just seen that, in regions of Gamma where sigma> 1, the trajectories for gamma = 1.1 exhibit sensitive phase dependence on initial data, as shown, for instance, by Fig. 3b, by the red dots in Fig. 4c and by the red curve in Fig. 4d. Sensitive dependence on initial data is usually associated with chaotic dynamics, but in this case the dynamics is periodic. This paradox is resolved by noting that such sensitivity concerns only the phase of the periodic trajectories, as already noticed in the previous subsection and, in addition, it occurs only if the initial data lie outside the attractor, e.g, elsewhere on Gamma; on the attractor, this phase sensitivity disappears, as we will show below. On the contrary, in the chaotic case gamma = 1.35, the sensitivity to initial data for trajectories with sigma > 1 always holds, off the attractor as well as on it, in excellent agreement with the chaotic character of the dynamics in the latter case."

In any case, in the same section we have clearly explained the fundamental difference between this sensitivity and that associated with chaotic dynamics, so we believe there can be no misunderstanding.

Having said this, we believe that this dynamical behavior deserves to be highlighted. Moreover, at the end of section 4.1 we have pointed out this: "... In our nonautonomous system, the amplitude of the periodic forcing epsilon plays a similar role. This transition to chaos induced by time-dependent forcing appears, therefore, to be directly linked to the existence of regions in phase space in which sensitive dependence to initial data occurs in the limit of periodic solutions. Thus, the chaotic behavior merely due to the time-dependent nature of the forcing can be traced back to the apparently paradoxical property of the autonomous system that was emphasized in Sect. 3.2 This striking observation deserves to be analyzed in greater depth in future studies. ..." We have therefore not removed section 3.2.

————————————————————————-

REVIEWER: Pg 5, line 4: please use the term " time evolution" instead of "time series", as in Pierini 2016, throughout the text.

RESPONSE: Done. Thank you.

————————————————————————-

REVIEWER: Pg 5 line 7: (X,Y) is not defined. Is it the initial condition of (Psi1,Psi3) as in eq. 5?

RESPONSE: Yes, but as you noticed, the definition comes later. Thank you very much. In the revised version, we write "depending on the initial point" (p. 5, l. 11).

————————————————————————-

REVIEWER: Pg 6 line 5: why sigma> 1 initially? sigma= 1 initially by definition (5).

RESPONSE: The reviewer is absolutely right. Thank you very much. We have removed "initially".

————————————————————————-

REVIEWER: Pg 7, Figure 3c. with gamma=1.35 you get chaos. How is it possible that two initially close trajectories never diverge?

RESPONSE: Yes, the red and blue curves are virtually coincident; this is what happens for trajectories leaving from the very restricted cold-color regions of Fig. 2d; on the other hand, this possibility is not surprising in view of the results of Pierini et al. (2016). Of course, these trajectories are nonetheless unstable once they have converged onto the PBA because, as noted in section 4.2 "... they will always pass sufficiently near trajectories that are chaotic, thanks to the mixing properties of the [PBA].".

REVIEWER: This contradicts the statement at the beginning of pg 9.

RESPONSE: We thank the reviewer for this comment. Now we have specified that we

are referring to cases with sigma>1 (p. 9, l. 18-20): "On the contrary, in the chaotic case gamma = 1.35, the sensitivity to initial data for trajectories with sigma>1 always holds, off the attractor as well as on it, in excellent agreement with the chaotic character of the dynamics in the latter case."

——————————————————————————-

REVIEWER: Pg 9, line 7 and following lines. I understand that you cannot compute with enough accuracy the first Lyapunov exponent (why? I found it strange for such a low dimensional system). If so, you cannot say that "the results prove unequivocally the assumption.."

RESPONSE: We thank the reviewer for this comment. Now we have changed "prove unequivocally" with "are consistent with" (p. 9., l. 25). In the following lines, we explain the typical problems encountered in the computation of the finite-time Lyapunov exponents, but we have not implied that it is impossible to compute them with enough accuracy.

——————————————————————————-

REVIEWER: Pg 9 line 11: illustrated instead of summarized.

RESPONSE: Done. Thank you.

——————————————————————————-

REVIEWER: Pg 9 caption of figure 5 and in other parts of the text: "non chaotic".

RESPONSE: Done. Thank you.

——————————————————————————-

REVIEWER: Pg 9 line 17: 4TDelta=100

RESPONSE: Done. Thank you.

——————————————————————————-

REVIEWER: Pg 10 line 25. It is a contradictory statement. Please explain it better.

RESPONSE: We thank the reviewer for this comment. Please see p.11, l. 11-13: "Recall that the PBAs of a periodically forced system (e.g., Pierini, 2014, and references therein) are always periodic. This periodicity of the PBAs occurs for both periodic and chaotic systems; the latter are typically referred to as cyclostationary. For the sake of simplicity we will refer below to chaotic and non chaotic PBAs, abbreviated as CPBAs and NPBAs, respectively."

————————————————————————-

REVIEWER: Pg 15 eq 10: I do not see the reason to introduce a new symbol for Cmax. There are already a lot of symbols to keep in mind.

RESPONSE: We thank the reviewer for this comment. We have removed the symbol $c_{max}$ and left only Theta; see the new Eq. (9).

————————————————————————-

REVIEWER: Pg18, last line: <Theta> -> <Theta>Gamma

RESPONSE: Done. Thank you.

————————————————————————-
————————————————————

**Fig. 1.** Figure 4 (modified)

[Figure]

(a): Return map of the minima

(b): Return map of the minima

**Fig. 2.** Figure A3 (new)

[Figure]

---

## Author Response (AR2)

**Review by Editor Juan Manuel Lopez**

We would like to thank the Editor for having drawn our attention to two issues raised by Referee 1. Please find below our responses to those comments.

————————————————————————-

5  EDITOR / REVIEWER 1:

1) Chaotic attractors of periodically driven systems can be observed on stroboscopic maps taken with the period of the driving, on which the dynamics appears to be autonomous. For this reason, these atttractors are usually seen as "normal" attractors, i.e. not pullback attractors in classical texts. Since most of the paper deals with a periodically driven system, the question that arises naturally is whether we are really observing pullback attractors, as the Referee is clearly asking in his/her report. This is a question that would rise in most readers mind with some knowledge on the pullback attractors issue and may need to be clarified.

Therefore, I suggest the authors to discuss this point early on in their manuscript. I wonder whether a citation of the mathematical proof that a general class of driven systems, of which the model used here is an example, given in the Appendix of Pierini, S., Ghil, M., and Chekroun, M. D.: Exploring the pullback attractors of a low-order quasigeostrophic ocean model: the deterministic case, J. Climate, 29, 4185–4202, 2016 would help clarifying the issue.

RESPONSE:

We thank the Editor for drawing our attention to the valuable comment of Referee 1 that we had overlooked in our first revision. Please see the new discussion that now appears in section 4 (page 11, lines 10-23).

————————————————————————-

20  EDITOR / REVIEWER 1:

2) Referee 1 report also mentions that the authors use ensembles instead of long time series, both methods are known to be equivalent in cases with constant or time-periodic driving. However, in Sec 5.2 a stochastically driven, so aperidically forced, system is briefly analyzed, presumably using the same ensembles technique. Some clarification is needed to whether the equivalence of both methods still holds, in my opinion. Is this true for any time-dependent forcing?

Therefore, I suggest the authors to deal with these questions in their resubmission.

RESPONSE:

We thank the Editor for drawing our attention to the valuable comment of Referee 1 that we had overlooked in our first revision. Please see the new discussion that now appears in section 4.1 (page 12, lines 4-14).

————————————————————————-

30  EDITOR:

Please, correct the axes label of several figures where the text "Psi" instead of the Greek letter appears.

RESPONSE:

We have corrected the axis labels wherever appropriate. Thank you very much.

————————————————————————-

[revised manuscript text omitted]